# Different lanthanide elements induce strong gene expression changes in a lanthanide-accumulating methylotroph

Linda Gorniak,[1] Julia Bechwar,[1] Martin Westermann,[2] Frank Steiniger,[2] Carl-Eric Wegner[1]

**ABSTRACT** Lanthanides (Ln) are the most recently described life metals and are central to methylotrophy (type of metabolism in which organic substrates without carbon-carbon bonds serve as carbon and energy source) in diverse taxa. We recently characterized a novel, Ln-dependent, and Ln-accumulating methylotroph, Beijerinckiaceae bacterium RH AL1, which requires lighter Ln (La, Ce, Nd) for methanol oxidation. Starting from two sets of incubations, one with different La concentrations (50 nM and 1 µM) and one with different Ln elements [La, Nd, or an Ln cocktail (containing Ce, Nd, Dy, Ho, Er, Yb)], we could show that La concentration and different Ln elements strongly affect gene expression and intracellular Ln accumulation. Differential gene expression analysis revealed that up to 41% of the encoded genes were differentially expressed. The effects of La concentration and Ln elements were not limited to Ln-dependent methanol oxidation but reached into many aspects of metabolism. We observed that Ln influence the flagellar and chemotactic machinery and that they affect polyhydroxyalkanoate biosynthesis. The most differentially expressed genes included *lanM*, coding for the well-characterized lanthanide-binding protein lanmodulin, and a glucose dehydrogenase gene linked to the conversion of β-D-glucose to D-glucono-1,5-lactone, a known potential metal chelator. Electron microscopy, together with RNAseq, suggested that Beijerinckiaceae bacterium RH AL1 can discriminate between Ln elements and that they are differently taken up and accumulated. The discrimination of Ln and links between Ln and various aspects of metabolism underline a broader physiological role for Ln in Beijerinckiaceae bacterium RH AL1.

**IMPORTANCE** Since its discovery, Ln-dependent metabolism in bacteria attracted a lot of attention due to its bio-metallurgical application potential regarding Ln recycling and circular economy. The physiological role of Ln is mostly studied dependent on presence and absence. Comparisons of how different (utilizable) Ln affect metabolism have rarely been done. We noticed unexpectedly pronounced changes in gene expression caused by different Ln supplementation. Our research suggests that strain RH AL1 distinguishes different Ln elements and that the effect of Ln reaches into many aspects of metabolism, for instance, chemotaxis, motility, and polyhydroxyalkanoate metabolism. Our findings regarding Ln accumulation suggest a distinction between individual Ln elements and provide insights relating to intracellular Ln homeostasis. Understanding comprehensively how microbes distinguish and handle different Ln elements is key for turning knowledge into application regarding Ln-centered biometallurgy.

**KEYWORDS** lanthanides, lanthanome, RNAseq, EDX, TEM, FFTEM, methylotrophy

Address correspondence to Carl-Eric Wegner, carl-eric.wegner@uni-jena.de.

The authors declare no conflict of interest.

See the funding table on p. 17.

Lanthanides (Ln) (Table S1) have been dubbed vitamins of the 21st century due to their relevance for high-tech applications central to our modern life (1). In nature, Ln primarily occur as poorly soluble hydroxides, carbonates, and phosphates (2–5), making accessing and recovering them challenging. Ln are also "life metals" (6–8) with high

relevance for carbon cycling and methylotrophs, microorganisms that utilize reduced carbon substrates without carbon-carbon bonds, such as methanol, as carbon and energy source. Ln-dependent metabolism is centered around pyrroloquinoline quinone (PQQ)-dependent alcohol dehydrogenases (ADHs). The catalytic activity of PQQ ADHs is based on a metal:PQQ cofactor complex. PQQ ADHs are diverse enzymes, including, among others, Mxa-type methanol dehydrogenases (MDHs) and five clades of Xox-type MDHs (9). Mxa-type, calcium-dependent MDHs were previously considered essential for microbes utilizing $C_1$ substrates such as methane or methanol (10–12). Xox-type MDH from *Methylorubrum extorquens* AM1 was the first identified and characterized Ln-dependent enzyme (13).

Genes encoding Xox-type MDH are widely distributed in the environment (14–17), suggesting that methylovory (the supplemental use of $C_1$ compounds as energy sources) is rather common (18–20). ExaF from *M. extorquens* AM1 was the first known Ln-dependent PQQ ADH acting on multicarbon substrates (21). Related enzymes have been identified, for instance, in the non-methylotroph *Pseudomonas alloputida* KT2440 (22) and the facultative methylotroph Beijerinckiaceae bacterium RH AL1 (23). Characteristic amino acid residues involved in Ln coordination indicate that most PQQ ADHs are Ln dependent (9, 24); the substrate spectrum of most of these enzymes is unknown.

Methylotrophs studied have a preference for lighter Ln (La-Nd). Heavier Ln are generally less favored or not utilized (6, 7). Ln-utilizing microbes must be able to mobilize and take up Ln, despite potentially low bioavailability. For *M. extorquens*, it was shown that Ln uptake is enabled by a transport system comprising a TonB-dependent receptor (LutH) and an ABC transporter (LutAEF), which are encoded in the *lut*-cluster (<u>l</u>anthanide <u>u</u>tilization and <u>t</u>ransport) (25). LutH is responsible for periplasmic uptake, while LutAEF facilitates cytoplasmic uptake. The first identified and best-studied Ln-binding protein, besides PQQ ADHs, was lanmodulin (LanM). LanM is a homolog of the well-characterized calcium-binding protein calmodulin (26), which features high affinities for Ln and application potential for lanthanide detection and recovery (27–29). Intracellular accumulation of Ln was shown for *M. extorquens* AM1 (25) and Beijerinckiaceae bacterium RH AL1 (30). *M. extorquens* stores Ln in the cytoplasm, while strain RH AL1 keeps periplasmic Ln deposits. RNAseq analyses of *M. extorquens* grown with soluble and less soluble Ln led to the identification of a gene cluster linked to the biosynthesis of a Ln chelator ("lanthanophore") (31).

In this study, we used Beijerinckiaceae bacterium RH AL1 (23, 30) grown with methanol as the carbon source, to study the effects of different La concentrations and different Ln elements on overall gene expression and intracellular Ln accumulation through RNAseq and electron microscopy. We were in particular interested in how far Ln reach into metabolism beyond the Ln-dependent methanol oxidation machinery and if different Ln elements change gene expression differently. We found that up to 41% of the encoded genes were differentially expressed when La was swapped for Nd or a pooled cocktail of light and heavy Ln (Ce, Nd, Dy, Ho, Er, Yb). Electron microscopy showed that strain RH AL1 accumulates Nd, as shown before for La (30), in the periplasm. Periplasmic storage was also visible for the Ln cocktail. Ln elements were differently accumulated, supporting the idea of preferential Ln uptake (30). We could show that La concentration and different Ln elements affected many different metabolic aspects on gene expression level. These included chemotaxis and motility, as well as polyhydroxyalkanoate metabolism, which are linked or controlled by Ca. We hypothesize that Ln partially interfere with or complement the physiological role of Ca.

## RESULTS

### Lanthanum concentration, lanthanide elements, and their effects on growth

We carried out two sets of incubations with Beijerinckiaceae bacterium RH AL1 and methanol (0.5%, vol/vol, 123 mM) as the carbon source (Fig. 1). For the first one, we used two different La concentrations (50 nM and 1 µM); for the second, we altered the added Ln elements and used La (1 µM), Nd (1 µM), or an Ln cocktail (Ce, Nd, Dy, Ho, Er, Yb;

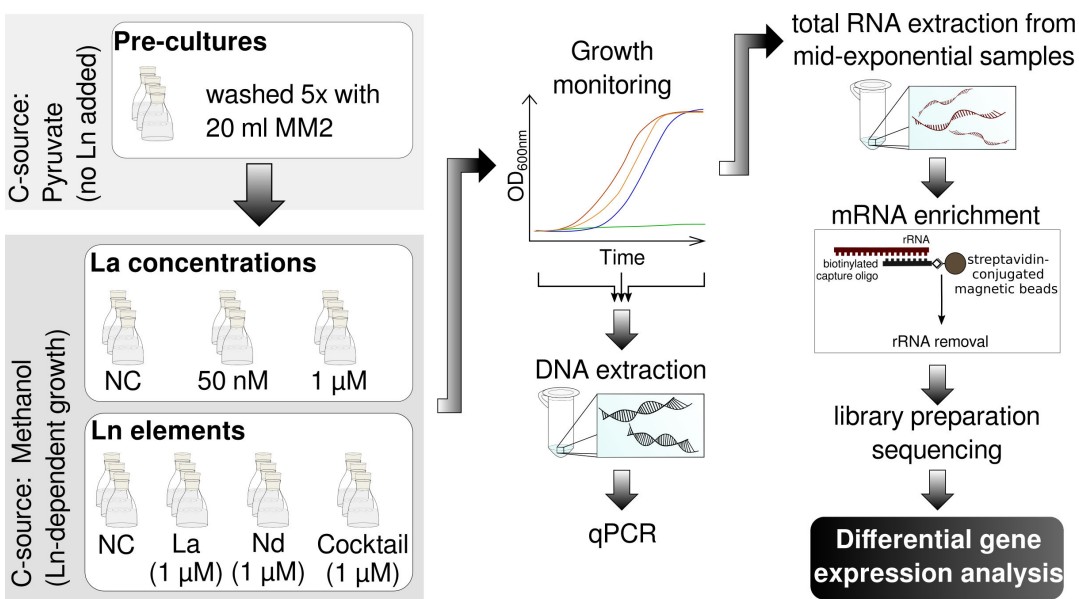

FIG 1 Overview of the cultivation setup and workflow for the carried out RNAseq experiments. Pre-cultures of Beijerinckiaceae bacterium RH AL1 were grown with pyruvate (0.2%, wt/vol, 18.175 mM) and washed with basal MM2 before being used as inoculum for two sets of incubations. Methanol (0.5%, vol/vol, 123 mM) was used as the carbon source for both sets, one investigating the effect of different (i) La concentrations (50 nM vs 1 µM), and one (ii), the effect of different Ln elements [La vs Nd vs Ln cocktail (Ce, Nd, Dy, Ho, Er, Yb)]. Cultivations were performed in triplicates ($n = 3$). Medium MM2 supplemented with methanol but without Ln source served as negative control (NC). Samples for DNA extraction and downstream quantitative PCR (qPCR) to determine cell numbers based on *lanM* gene copies were taken at the beginning and end of the incubation and during mid- to late-exponential phase. Biomass samples for RNA extraction were taken during mid- to late-exponential growth as well. Total RNA from each biological triplicate was enriched for mRNA by means of subtractive hybridization, before being subjected to library preparation, and Illumina sequencing. Pre-processed sequencing data were the starting point for differential gene expression analysis.

pooled, 0.9 µM combined Ln, the exact composition is given in Table S2). Strain RH AL1 depends on Ln for growth with methanol, and we observed previously intracellular La accumulation (30). We cultivated pre-cultures with pyruvate (0.2%, wt/vol, 18.175 mM) as an alternative carbon source to avoid Ln carryover.

We noted significantly ($P \leq 0.05$, Student's *t*-test) different growth rates for 50 nM and 1 µM La cultures ($0.038 \pm 2.13 \times 10^{-4}$ h$^{-1}$ and $0.044 \pm 0.001$ h$^{-1}$) (Fig. 2A, left panel; Table S3). Cell numbers increased from $2.22 \pm 3.79 \times 10^{6}$ and $2.47 \pm 3.49 \times 10^{6}$ mL$^{-1}$ (t$_0$, start of the incubation) to $1.49 \pm 0.71 \times 10^{9}$ and $2.64 \pm 0.51 \times 10^{9}$ mL$^{-1}$ (t$_2$, end of the incubation) (Fig. 2B, left panel; Table S4). The incubations with different Ln elements revealed overall comparable growth patterns (Fig. 2A, right panel), but cultures grown with La showed, compared to Nd cultures, significantly ($P \leq 0.05$) faster growth with a doubling time of $16.70 \pm 0.46$ h and a growth rate of 0.041 (h$^{-1}$). Cell numbers increased up to $2.41$–$3.68 \times 10^{9}$ mL$^{-1}$ (t$_2$) for the different setups (Fig. 2B, right panel; Table S4).

## Overall transcriptome changes

Biomass samples for RNAseq were taken during the mid- to late-exponential phase (t$_1$, Fig. 2A) after the cultures reached OD$_{600nm}$ values between 0.243 and 0.489. Genes with changes in the expression above |0.58| log$_2$ fold change (log$_2$FC) (equivalent to a 50% change in gene expression) and expression values higher than 4 log$_2$ counts per million (log$_2$CPM) (Fig. S1 to S4; Table S5) were considered for differential gene expression analysis (DGEA) (Fig. 3A; Tables S6 to S9). We made the following comparisons: (1) 50 nM La vs 1 µM, (2) La vs 1 Nd, (3) La vs Ln cocktail, and (4) Nd vs Ln cocktail (Fig. 3A). Up to 41% of the encoded genes were differentially expressed in the case of 2 and 3 (Fig. 3B). Gene expression differed less for 4 and 1. We identified 320 and 351 differentially expressed genes (DEGs), representing 7.4% and 8.1% of the encoded genes, respectively.

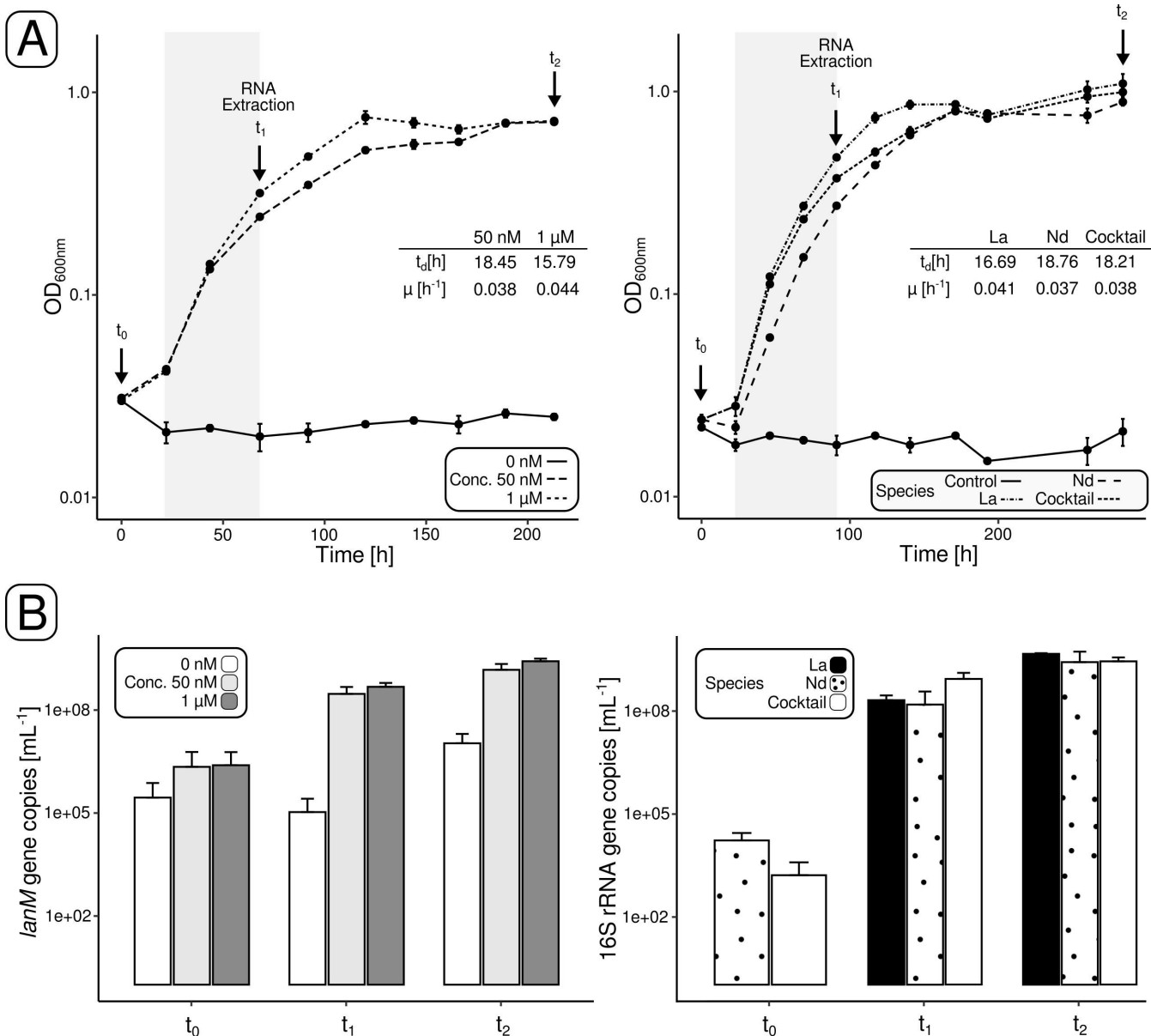

**FIG 2** Methylotrophic growth of Beijerinckiaceae bacterium RH AL1 with methanol as the carbon source (0.5%, vol/vol, 123 mM) and either different concentrations of La or different Ln elements. (A) Growth curves of strain RH AL1 grown with either two different concentrations (50 nM and 1 µM) of La or different Ln elements [1 µM of either La or Nd or 0.9 µM of a Ln cocktail (Ce, Nd, Dy, Ho, Er, Yb)]. Cultures without added Ln served as negative control. Cultivations were performed in triplicates ($n = 3$). Growth was monitored spectrophotometrically ($OD_{600nm}$). The gray shade indicates the time interval considered for calculating doubling time ($t_d$) and growth rate ($\mu$). Samples for molecular work were taken at time points $t_0$–$t_2$ including samples for RNA extraction ($T_1$). (B) Growth monitoring by *lanM* gene-targeting quantitative PCR. Samples for DNA extraction and qPCR were taken at the beginning ($T_0$), the mid- to late-exponential ($T_1$), and stationary phase ($T_2$).

Taking a closer look at the DEGs in strain RH AL1 revealed a substantial overlap of 1,165 genes between 2 and 3 (Fig. 3C). Independent of the comparison, a subset of 105 genes was differentially expressed in all cases. Overrepresentation analysis based on KEGG annotations revealed an enrichment of genes linked to the KEGG pathways two-component systems, flagellar assembly, and chemotaxis (Fig. 3D; Table S10). The latter two partially overlap with two-component systems. We identified two sensor kinase genes coding for PleC, linked to cell cycle progression, pole morphogenesis, and cell envelope integrity and CheA associated with chemotaxis (32). KEGG modules

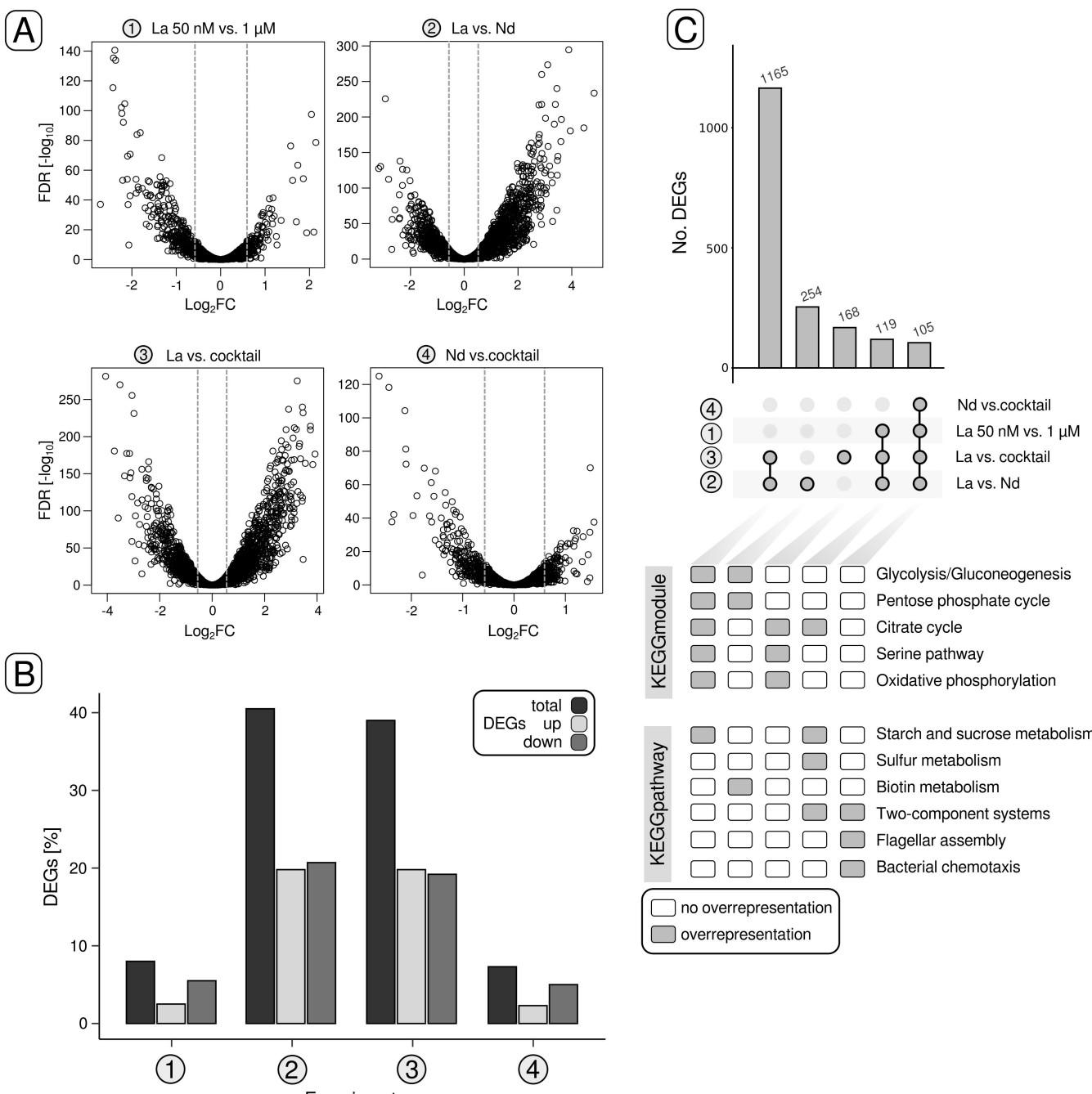

FIG 3 Differential gene expression in response to La concentration and Ln elements. (A) Volcano plots indicate differentially expressed genes (DEGs) for the four different comparisons 1–4. Genes with changes in gene expression above |0.58| log₂FC, gene expression values higher than 4 (log₂CPM), and P-values smaller than 0.05 were considered for downstream analysis. The gray dashed lines indicate the log₂FC threshold of |0.58|. (B) The proportions of total DEGs, upregulated genes, and downregulated genes in relation to the number of genes encoded in the genome are shown as a bar chart. (C) UpSet plots were used to highlight sets of genes (>100 genes) that were either shared between different comparisons or unique. The lower half of (C) indicates potential overrepresentation of KEGG modules and KEGG pathways. The categories listed under KEGG modules include partially multiple KEGG modules: glycolysis/gluconeogenesis (M00001, M00003), pentose phosphate cycle (M00004), citrate cycle (M00009, M00011, M00173), serine pathway (M00346), and oxidative phosphorylation (M00144). FDR, false discovery rate.

related to central carbohydrate and energy metabolism were overrepresented in the 1,165 overlapping genes between 2 and 3.

## Differentially expressed genes and pathways

Based on DGEA, numerous aspects of metabolism were affected by applying different La concentrations and Ln elements. Motility- and chemotaxis-related genes were downregulated in all comparisons (Fig. 4; Tables S6 to S9 and Table S11). In the case of 2 and 3, $log_2FC$ values for flagellar genes ranged from −0.60 to −4.27 ($log_2CPM$ values between 4.25 and 9.33). We investigated the effect of Ln concentration (5 nM to 10 µM) and elements (La, Nd, Ln cocktail) on motility by employing a soft agar and 2,3,5-triphenyl-tetrazolium chloride (TTC)-based assay (Fig. S5; Table S12). We observed a decrease in motility with increasing Ln concentration. Motility was higher for cultures grown with Nd or the Ln cocktail than La.

Genes encoding proteins enabling alkanesulfonate uptake and utilization (SsuABCD) were downregulated when strain RH AL1 was grown with 1 µM ($log_2FC$ −0.59 and −0.84, $log_2CPM$ 4.25–7.40) instead of 50 nM La but upregulated in Nd and Ln cocktail samples ($log_2FC$ 1.17–3.50) (Table S6-8, and S11). We noticed numerous genes associated with

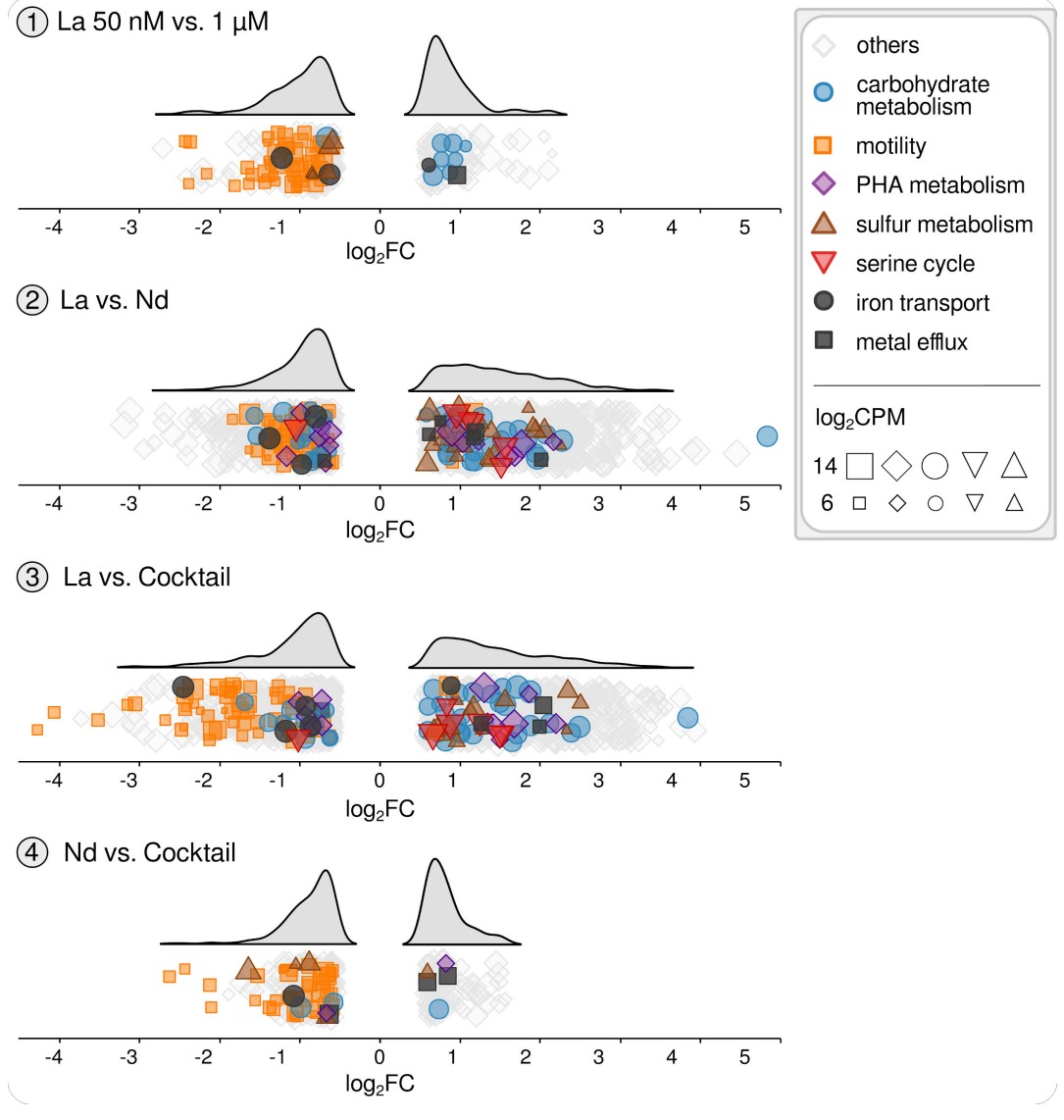

**FIG 4** Differential gene expression relating to selected metabolic aspects in response to different La concentrations and Ln elements. The responsive genes are categorized and listed in Table S11. For each comparison, the up- and downregulation of genes are illustrated through ridge plots based on $log_2FC$ values. Marker sizes correspond to the $log_2CPM$ values. Selected genes associated with motility, carbohydrate metabolism, PHA metabolism, serine cycle, sulfur metabolism, iron transport, and metal efflux were highlighted. PHA, polyhydroxyalkanoate; $log_2CPM$, $log_2$ counts per million.

heavy metal efflux, iron transport, and polysaccharide export, as well as genes for porins (Table S11). The latter were mostly downregulated upon increased La concentration and when La was swapped for Nd or the Ln cocktail ($\log_2$FC between −0.60 and −2.06). Genes coding for the iron-storage protein bacterioferritin (RHAL1_00987) and EfeUO (RHAL1_03444, RHAL1_03445), part of the EfeUOB iron uptake system, were downregulated in the case of 2 and 3 ($\log_2$FC between −0.89 and −1.17). The same was true for the *exbB* and *exbD* genes that encode the energy transmission machinery for TonB-dependent uptake ($\log_2$FC between −0.82 and −1.40) in the case of Nd.

A gene (RHAL1_01212) encoding a glucose-1 dehydrogenase was the (second) most upregulated ($\log_2$FC 4.83 and 3.84, $\log_2$CPM 10.08) in the case of 2 and 3. Genes scattered across central carbohydrate metabolism pathways were differentially expressed and predominantly upregulated (Fig. 4; Tables S7 and S8) in response to Nd and the Ln cocktail, including succinate dehydrogenase (flavoprotein subunit), malate dehydrogenase, and fumarate hydratase genes ($\log_2$FC 0.76–1.14). Formate dehydrogenase genes were slightly upregulated with increased La concentration (RHAL1_03901–03903, $\log_2$FC 0.72–0.88). Gene expression linked to $C_1$ assimilation through the serine cycle was affected by swapping the Ln elements. The gene encoding serine hydroxymethyltransferase (RHAL1_01794) was upregulated when strain RH AL1 was grown with either Nd or the Ln cocktail ($\log_2$FC 0.95 and 0.88); *sga* (serine-glyoxylate aminotransferase, RHAL1_01353) was higher expressed when strain RH AL1 was supplemented with the Ln cocktail ($\log_2$FC 0.66). At the same time, *hpr* (hydroxypyruvate reductase, RHAL1_03822) was less expressed ($\log_2$FC −1.02).

Strain RH AL1 features polyhydroxybutyrate [PHB, a common polyhydroxyalkanoate (PHA)] vacuoles at its cell poles (23), a common characteristic of members of the family Beijerinckiaceae (33). Cultivation with Nd and the Ln cocktail caused differential expression of PHA (de)polymerization-related genes. A gene encoding the PHA synthesis repressor PhaR (RHAL1_03606) was downregulated in Ln and Nd samples ($\log_2$FC −1.16 and −1.02). Two genes (RHAL1_01143 and RHAL1_3443), coding for phasins (proteins with a scaffolding role in PHA granule formation), were highly expressed and upregulated in Nd and Ln cocktail cultures ($\log_2$FC 0.90 and 1.76, $\log_2$CPM 12.22 and 13.62). Strain RH AL1 carries four PHA depolymerase genes. Two were upregulated (RHAL1_03589, RHAL1_01066), and two were downregulated (RHAL1_01171, RHAL1_01370), with $\log_2$FC values ranging from −0.68 to 2.19 ($\log_2$CPM between 4.97 and 8.65).

## The lanthanome in response to lanthanide elements and concentration

The lanthanome comprises all biomolecules, especially proteins, partaking in Ln utilization (27). We took a closer look at four groups of lanthanome-related genes: the lanthanome core, PQQ biosynthesis-related genes, and genes of the *LC*- and *lut*-cluster (Fig. 5). We define the core lanthanome here as genes coding for key proteins involved in Ln-dependent methanol oxidation. *Lut*- and *LC*-clusters refer to identified gene homologs of the previously described *lut*- (Ln utilization and transport) and *LC*-clusters (Ln chelator) (25, 31). Central to methanol oxidation in Beijerinckiaceae bacterium RH AL1 is one clade 5 XoxF MDH. The *xoxF5* (RHAL1_02998) gene was moderately upregulated in response to Nd and the Ln cocktail ($\log_2$FC 1.13 and 1.04). Gene expression was high across all conditions ($\log_2$CPM 10.61–13.26) (Table S13). The same was true for the gene encoding XoxG (RHAL1_02256) ($\log_2$CPM 9.46–12.02), the complementary cytochrome $c_L$ of XoxF. Strain RH AL1 encodes two more PQQ ADHs, the broad substrate ADH ExaF and a subgroup 9 PQQ ADH. The expression of *exaF* (RHAL1_01345) did not differ significantly when comparing conditions. RHAL1_03347, encoding the subgroup 9 PQQ ADH, was slightly upregulated in response to Nd ($\log_2$FC 0.74). Differences in La concentration led to a strong downregulation ($\log_2$FC −2.71) of *lanM* (RHAL1_01396), encoding lanmodulin, while Nd and the Ln cocktail triggered upregulation ($\log_2$FC 3.29, 2.08). Comparing Nd and Ln cocktail incubations showed downregulation of *lanM* ($\log_2$FC −1.20) for the latter.

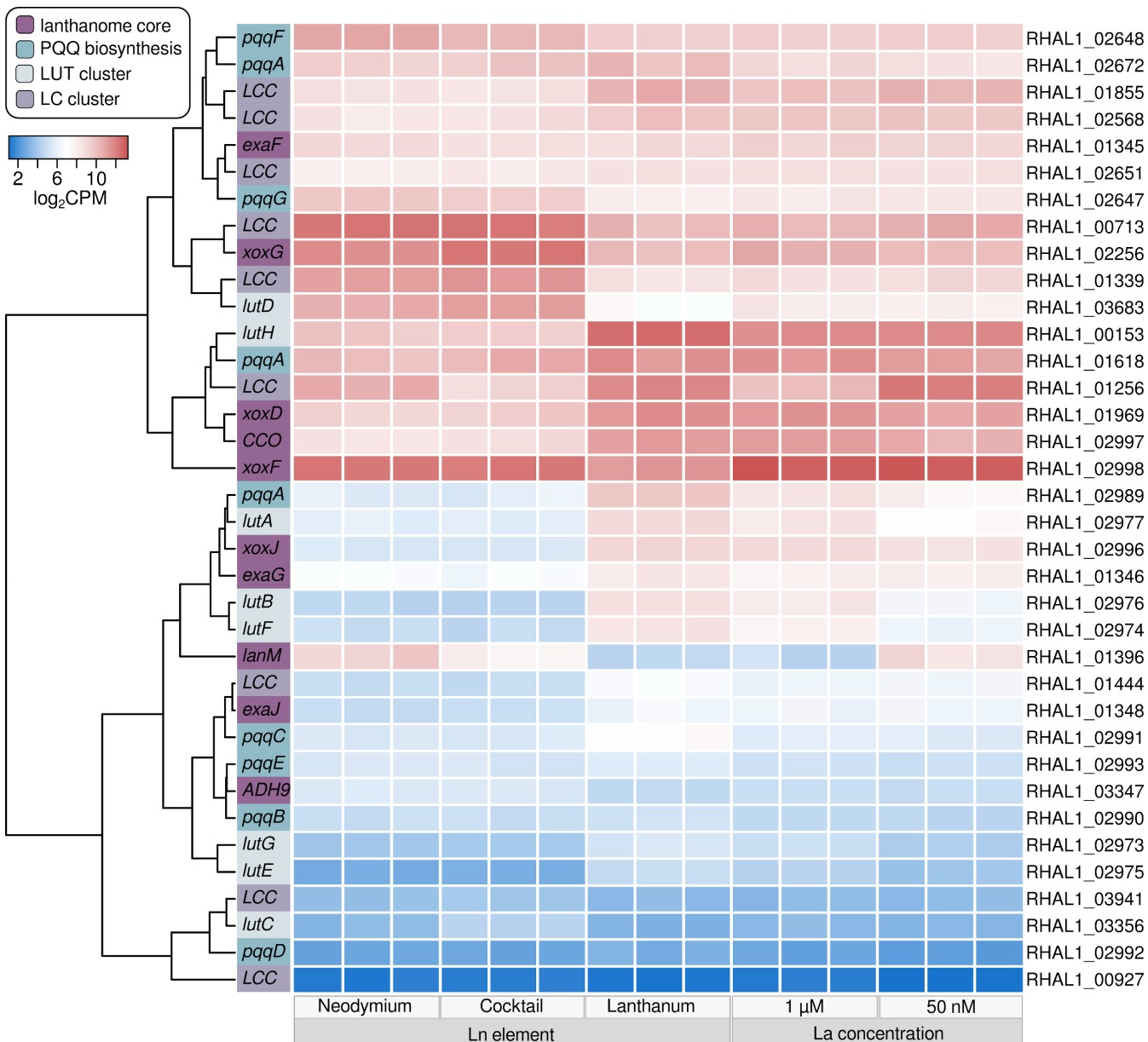

**FIG 5** Gene expression of lanthanome(-related) genes. We defined four groups of genes linked to the lanthanome: lanthanome core, PQQ biosynthesis, *LC*- and *lut*-clusters. The core lanthanome comprises genes for key proteins involved in lanthanide-dependent methanol oxidation. *Lut*- and *LC*-clusters refer to identified gene homologs of the previously described *lut*-cluster (lanthanide utilization and transport) and lanthanide chelator cluster (25, 31). The row dendrogram is based on euclidean distances. The colored sidebar indicates the affiliation of individual genes regarding the mentioned grouping. Gene expression is given in log$_2$CPM (counts per million), indicated by a blue-red color scale, and shown for the different biological replicates per condition (*n* = 3).

Most gene homologs (Fig. 5; Table S14) of the *lut*- and *LC*-clusters were downregulated when comparing Nd and the Ln cocktail to La (Table S7 and S8). This was most obvious for *lutH*, which encodes the TonB-dependent receptor involved in periplasmic Ln uptake (log$_2$FC −3.11, −3.34). The gene homologs (RHAL1_02974, RHAL1_02975, RHAL1_02977) of the ABC transporter LutAEF, crucial for cytoplasmic Ln uptake, were upregulated when strain RH AL1 was grown with 1 µM La in comparison to 50 nM (log$_2$FC 0.69–0.92). The gene (RHAL1_03683) coding for the periplasmic Ln-binding protein LutD was strongly upregulated in Nd and Ln cocktail samples (log$_2$FC 2.86 and 3.37). RHAL1_01256, the gene homolog (Table S6 to S9) of the TonB-dependent receptor

of the LC cluster, was downregulated across all comparisons ($\log_2$FC between $-1.31$ and $-2.77$).

The cofactor PQQ coordinates Ln ions in Ln-dependent PQQ ADH. Genes associated with PQQ biosynthesis were differentially expressed in response to Ln elements. The three copies of *pqqA* (RHAL1_01618, RHAL1_02672, RHAL1_02989) were downregulated in response to Nd and the Ln cocktail. The degree of downregulation differed [$\log_2$FC (RHAL_01618) $-1.43$, $-1.91$; (RHAL1_02672, only for Nd) $-0.68$ (RHAL1_02989), $-2.68$, $-2.60$]. Likewise, *pqqC* was also downregulated in Nd and Ln cocktail incubations, while *pqqFG* (RHAL1_02647, RHAL1_02648) was moderately upregulated ($\log_2$FC 0.84–1.42).

## Differences in intracellular lanthanide deposition

We have been previously able to show intracellular, periplasmic La deposition in *Beijerinckiaceae* bacterium RH AL1 (30). Using transmission electron microscopy (TEM), freeze-fracture TEM (FFTEM), and elemental analysis by means of energy-dispersive x-ray spectroscopy (EDX), we probed strain RH AL1 for intracellular Ln accumulation when grown on methanol as the carbon source with Nd or the Ln cocktail. FFTEM complements TEM and is useful for obtaining detailed structural views of cellular topography. Screening ultrathin sections from cultures grown with Nd or the Ln cocktail revealed peripheral, periplasmic deposits in the proximity of the cell poles and close to PHB vacuoles (Fig. 6A, upper left and upper middle panel). FFTEM confirmed the localization of these intracellular deposits (Fig. 6A, upper right panel). We revisited ultrathin sections from previous work (30) and compared them to ultrathin sections from cultures grown with Nd and the Ln cocktail to analyze the effect of swapping Ln elements on PHA biosynthesis (Fig. 6B). The average cell area when strain RH AL1 was grown with La was $1.09 \pm 0.31$ $\mu m^2$, while the respective values were significantly [$P \leq 0.05$, one-way analysis of variance (ANOVA), post-hoc Tukey-Kramer test] higher for Nd ($1.79 \pm 0.58$ $\mu m^2$, +64%) and the Ln cocktail ($1.74 \pm 0.53$ $\mu m^2$, +59%), respectively (Fig. 6B). PHB vacuoles occupied between $0.11 \pm 0.04$ (La) and $0.58 \pm 0.20$ (Nd) $\mu m^2$, which was equivalent to between $10.5\% \pm 4.9\%$ and $32.9\% \pm 5.2\%$ of the cell area. The areas occupied by PHB vacuoles were 3.13 and 2.43 times bigger for Nd and the Ln cocktail than for La (Fig. 6B).

We verified that the identified periplasmic deposits contained Ln through EDX starting from ultrathin sections and freeze-fracture replicas (Fig. 7). EDX is based on elements emitting characteristic x-rays upon x-ray excitation. Concerning the samples originating from Ln cocktail incubations (Fig. 7), distinct signals were detected for Ce, Nd, Dy, Ho, and Er, but only weakly for Yb. The share of the different Ln in the deposits (Table S15) ranged from 1.9% (Yb) to 32.3% (Ce). We repeated the incubations with the Ln cocktail (Ce, Nd, Dy, Ho, Er, Yb) and collected cell-free supernatant samples during the late-exponential phase ($OD_{600nm}$ values between 0.335 and 0.539) to check for Ln depletion during incubation. Subsequent elemental analysis through inductively coupled plasma mass spectrometry (ICP-MS) revealed the complete depletion of Ce and Nd, while 0.16% (Dy), 0.51% (Ho), 3.01% (Er), and 75.93% (Yb) of the initially added other Ln could be detected (Table S16).

It was previously reported that Ln are potentially stored intracellularly as Ln phosphates in *M. extorquens* AM1 (25), which prompted us to determine P:Ln ratios based on EDX data for the deposits that we have identified. The ratio between P and Ln was between 0 [$n = 3$, EM data from reference (30)] for La deposits and 12 for Nd ($n = 3$) and Ln deposits ($n = 4$) (Fig. S6; Table S15).

## DISCUSSION

Observed gene expression changes, especially in response to different Ln elements, indicated a broader role for Ln in cellular metabolism in *Beijerinckiaceae* bacterium RH AL1, beyond the lanthanome and methylotrophy (Fig. 8). Strain RH AL1 is apparently able to discriminate (light and utilizable) Ln elements. Past studies reported positive effects on growth, dependent on the presence or absence of Ln in (non-)methanotrophic methylotrophs that possess Ca- and Ln-dependent MDH (34–39). These cultivations were

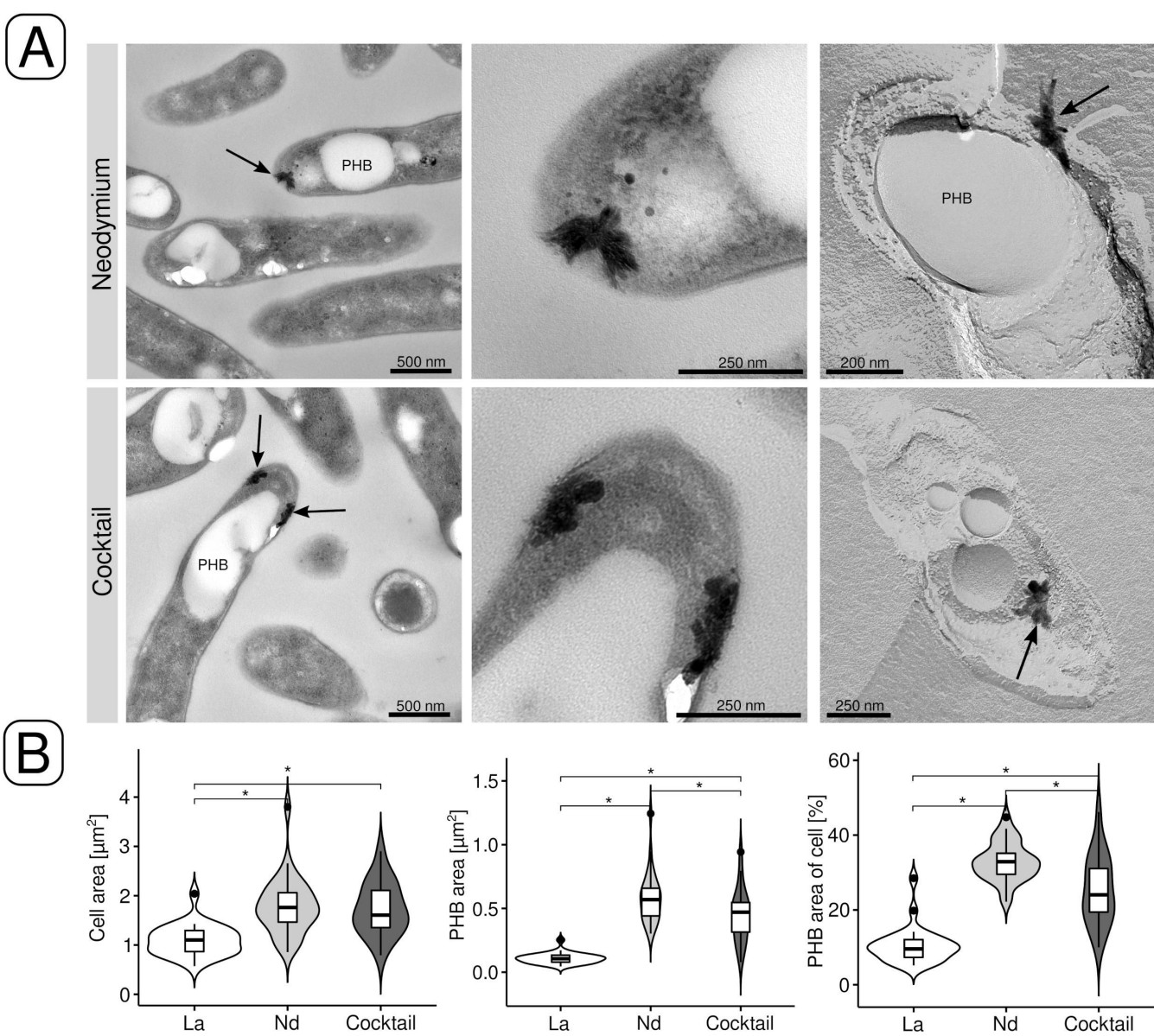

**FIG 6** Electron microscopic examination of periplasmic Ln deposits. (A) Deposits were identified and localized by TEM (left panels). The middle panels are close-ups of the areas indicated by black arrows. Periplasmic deposits were also localized and identified by FFTEM (right panels). (B) The size of cells grown with La (30), Nd, or the Ln cocktail was compared by measuring cell areas. We also compared the area occupied by polyhydroxybutyrate (PHB) vacuoles and the area of the cell occupied by them. The analysis was done for three images (magnification, 4,000×; image area, 540.5 µm$^2$), and between 27 and 30 cells were analyzed per condition (La, Nd, cocktail). Statistical significance was tested by means of one-way ANOVA, combined with a post-hoc Tukey-Kramer test. Asterisks indicate significant differences ($P \leq 0.05$).

typically performed using soluble Ln-chloride salts and concentrations between 10 µM and 30 µM (35, 37, 39). Few studies used in part lower Ln concentrations (38, 40). Ln concentrations in this study were lower compared to past studies and chosen based on the minimum (50 nM Ln) and optimum (1 µM) concentrations for strain RH AL1 when grown with methanol as the carbon source. (23). In the environment, Ln are not particularly rare. In the Earth' crust, lighter Ln (La, Ce) reach abundances of 60 and 120 ppm, comparable to common metals such as Cu and Zn (41). However, Ln bioavailability in soils is impaired by Ln being mostly present in poorly soluble mineral latices (42). The water soluble, readily bioavailable fraction of Ln makes up often less than 0.01% of the present Ln (43) in soils. Similar to previous results (23) and the findings presented here,

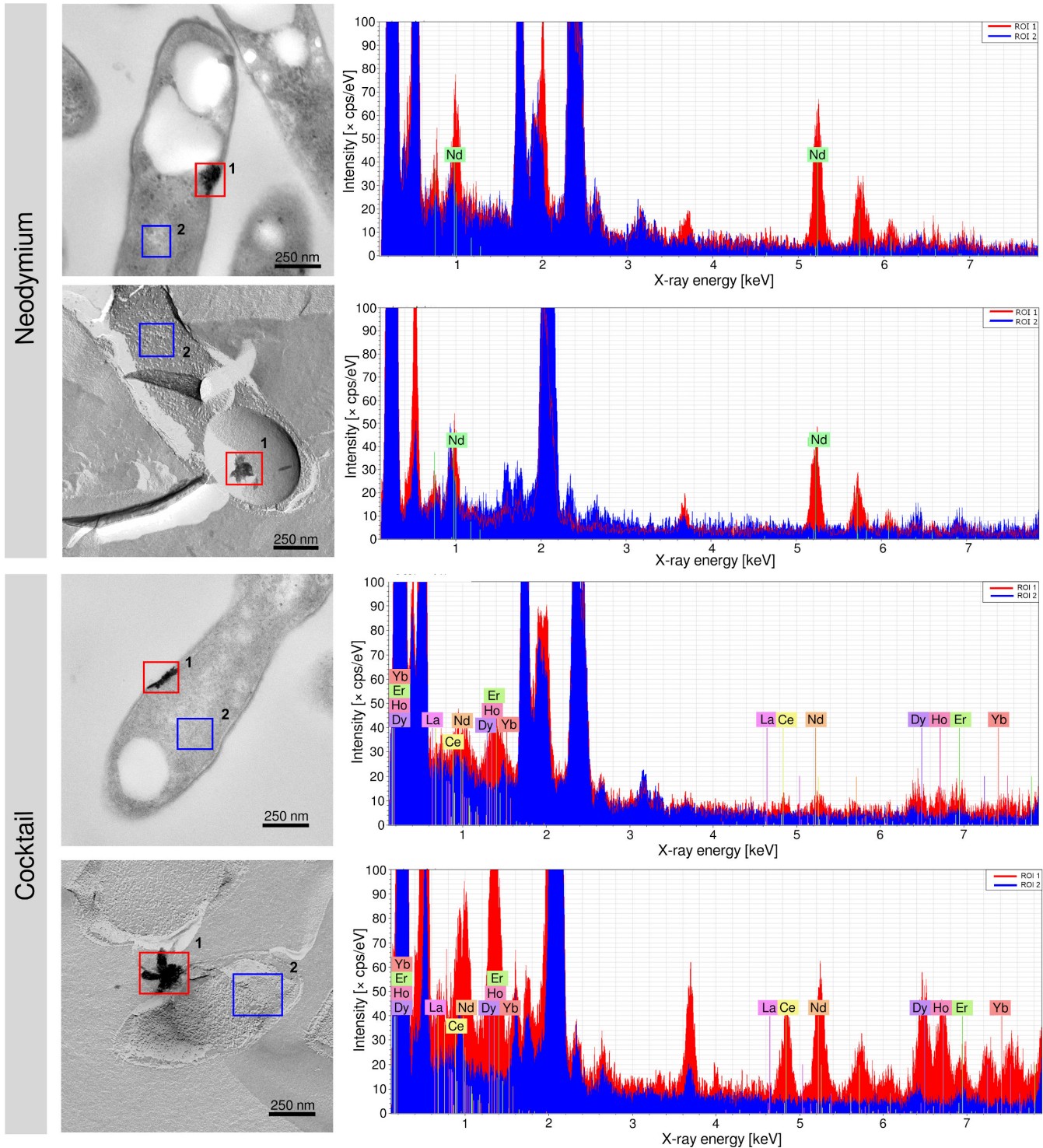

**FIG 7** Elemental analysis of periplasmic Ln deposits from incubations with Nd and the Ln cocktail. TEM and FFTEM specimens were subjected to elemental analysis based on EDX. Red (deposit) and blue (reference) boxes indicate measured areas. ROI, region of interest; cps, counts per second. Scale bar, 250 nm.

other studies noted only small differences when comparing the effect of different light Ln elements on growth (37, 40). Few studies previously addressed gene/protein expression changes in response to Ln supplementation (36–39) and only in microorganisms possessing Mxa- and Xox-type MDHs. In those, Ca- and Ln-dependent MDHs are inversely regulated, dependent on the presence of Ln, through a mechanism known as "Ln switch."

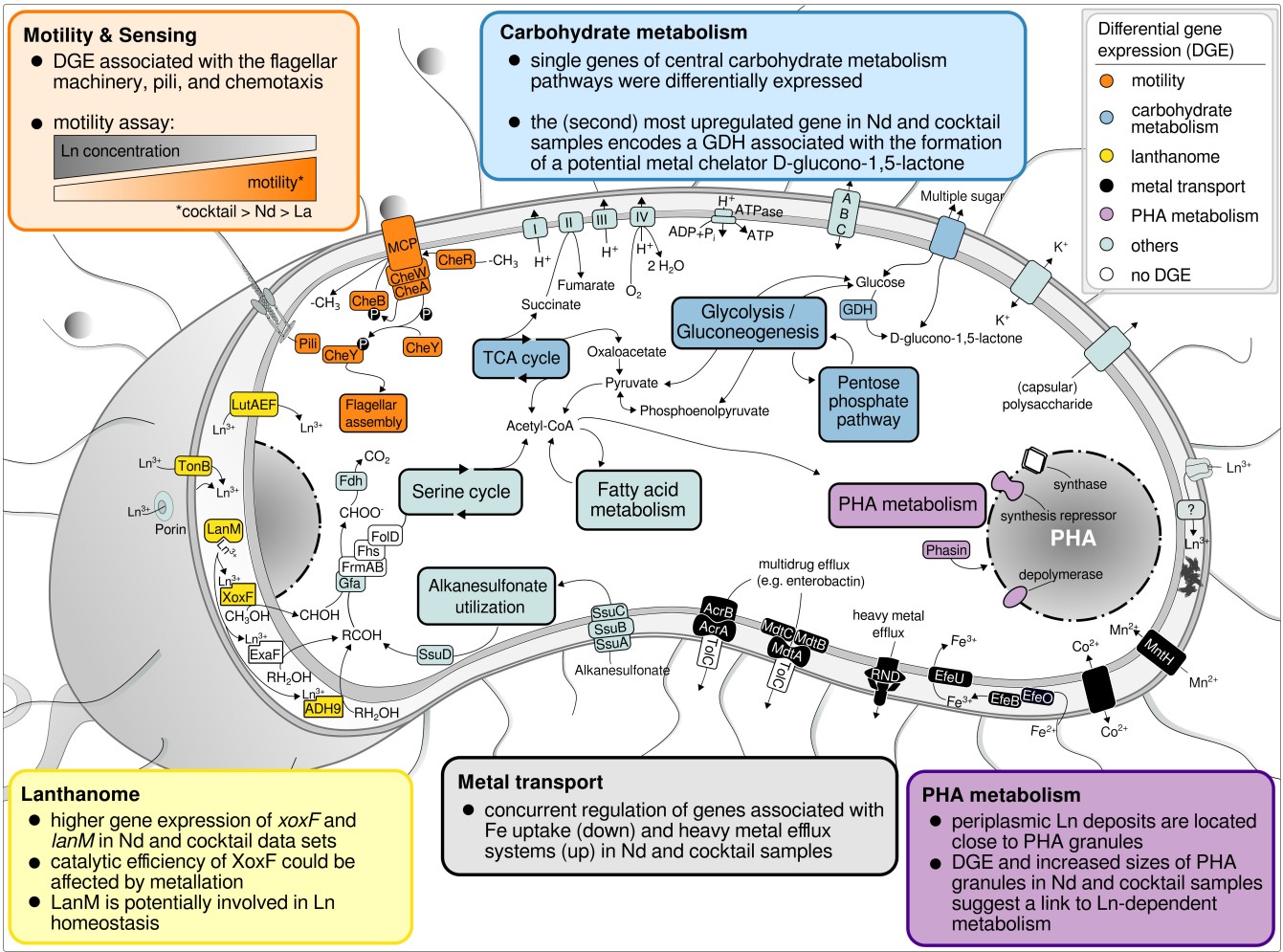

**FIG 8** Summary of metabolic aspects responding to differences in La concentration and added Ln elements in Beijerinckiaceae bacterium RH AL1 when grown with methanol as the carbon source. The color code of the text boxes refers to individual metabolic aspects. Colored proteins in the cell scheme highlight the differential expression of the genes encoding the corresponding protein. DGE, differential gene expression; GDH, glucose dehydrogenase.

Based on these data, it seemed as if Ln control rather small numbers of genes, mostly *mxa*- and *xox*-cluster genes. *Methylobacterium aquaticum* 22A is the only organism for which the effect of different Ln (La, Ho, Lu) on gene expression was tested. Only La affected the gene expression of methylotrophy-related genes (37).

Differences observed in our study among the La, Nd, and Ln cocktail treatments support that aspects of metabolism that use Ln are tuned toward certain Ln elements. In *M. buryatense* 5GB1C, La and Ce triggered the Ln switch, but La had a stronger effect on *xoxF* expression (44). Thermal stability analysis showed that XoxF metallation (defined as the acquisition of metals by proteins, here the experimental incorporation of different Ln elements) in *M. extorquens* AM1 affects the integrity and that La is preferred over Nd (45). The catalytic efficiency of XoxF was affected by metallation in *Methylacidiphium fumariolicum* SolV and higher for lighter Ln (46). The observed upregulation of *xoxF* and *xoxG* (encoding a $c_L$-type cytochrome) in the case of strain RH AL1 could have compensated for a potentially reduced catalytic efficiency with Nd. Differences in ionic radius, Lewis acidity, and redox potential likely determine the widespread preference for lighter Ln. The latter affects the physiological electron acceptor of XoxF, XoxG, which complements XoxF with a reduction potential tailored toward lanthanide elements that are ideally suited for XoxF (47).

It is known from iron homeostasis (48, 49) that TonB-ABC transport systems are commonly downregulated if iron is sufficiently available. The same was shown for *lutH* and *lutAEF*, which encode a TonB-ABC transport system for lanthanide uptake into peri- and cytoplasm (25, 50). We noted the same pattern in this work and also previously when strain RH AL1 was not grown with methanol but other carbon sources, whose utilization does not require Ln supplementation (30). The downregulation of PQQ biosynthesis genes when strain RH AL1 was grown with Nd and the Ln cocktail, especially *pqqA* copies, seems counterintuitive. XoxF requires PQQ as a cofactor, and *xoxF* was upregulated. Similar observations were made in *M. aquaticum* 22A when comparing methylotrophic growth with Ca and La (37).

Our findings suggest that Ln affect many aspects of metabolism (Fig. 8), including chemotaxis and motility, as well as PHA metabolism, which are known to be linked to Ca (calcium). The role of Ca as a regulator and secondary messenger is well established in eukaryotes (51–53) but poorly understood in prokaryotes (53, 54). Multiple aspects of physiology are assumed to be controlled by Ca in prokaryotes, including cell cycle progression, virulence, and competence (55–58). Ca modulates the phosphorylation state of the Che proteins, which are the basis for chemotaxis (54). Our gene expression data, supported by the carried out motility assays, support that Ln affect chemotaxis and motility by functioning as Ca analogs/mimics or antagonists. Our limited knowledge relating to Ca metabolism in prokaryotes includes its uptake. Complexes of short-chain PHB and polyphosphate (PP) form Ca channels and represent one important route for Ca uptake (59–63). Using patch-clamp techniques, La was found to compete with Ca for binding sites in these PHB-PP channels (64). Long chains of PHB, kept in a vacuole, are a common form of carbon storage. In previous work, we showed intracellular, periplasmic La accumulation in strain RH AL1 (30), primarily in close proximity to PHB vacuoles. We observed comparable periplasmic deposits when we grew strain RH AL1 with Nd or the Ln cocktail. We also noted differential expression of genes linked to PHA synthesis and PHA vacuole formation. An involvement of complexed PHB in selective Ln uptake (and storage) could explain the localization of the observed Ln deposits. Periplasmic Ca accumulation is a strategy for regulating intracellular Ca levels (65). Calmodulin is a well-known protein involved in intracellular Ca homeostasis in eukaryotes, which features multiple, calcium-binding EF-hand domains. Lanmodulin, a calmodulin-homolog with EF-hand domains tailored for Ln binding and first identified in *M. extorquens* AM1, was the first known Ln-binding protein (except PQQ ADHs). The responsiveness of *lanM* that we observed is not restricted to methylotrophic growth. We previously showed that *lanM* was the most differentially expressed gene in strain RH AL1 when comparing growth with pyruvate as the carbon source in the presence and absence of La (30). The exact role of lanmodulin is not clear yet. It is potentially involved in Ln storage, shuttling Ln to Ln-dependent enzymes, or intracellular homeostasis, like calmodulin in the case of calcium. We meanwhile know three more Ln-binding proteins: the periplasmic Ln-binding protein LutD, supposed to be associated with the LutAEF ABC transporter (27); a ubiquitin ligase from plant chloroplasts (66); and most recently lanpepsy (67) from *Methylobacillus flagellatus*. The gene encoding LutD in strain RH AL1 was among the most differentially expressed genes when exchanging La for Nd or the Ln cocktail. Its alleged association with the Ln-specific LutAEF ABC transporter (27) is a sign that it plays a role regarding cytoplasmic Ln uptake.

Beijerinckiaceae bacterium RH AL1 was the first Ln-utilizing bacterium shown to accumulate Ln in the periplasm (30) based on cultivations with La. Our EM data showed similar periplasmic deposits when strain RH AL1 was grown with Nd or the Ln cocktail, but observed Ln deposits differed in their elemental composition. For *M. extorquens* AM1, it was postulated that lanthanides are kept intracellularly in the cytoplasm complexed with polyphosphate (25). We did not detect P in periplasmic deposits from cultures grown with La. However, determined P to Ln ratios for Nd and Ln cocktail samples align with potential Ln phosphates. Differences in the composition of the observed Ln deposits suggest that different Ln elements are accumulated in distinct forms.

Periplasmic Ln deposition can contribute to Ln homeostasis. The deposits from Ln cocktail cultures contain mostly Ce ($32.26 \pm 4.47$ Ln%), different amounts (14–20 Ln%) of Nd, Dy, Ho, and Er; but hardly any Yb (~2 Ln%). Differences in the proportion of Ln elements in intracellular deposits as well as differences in depletion in spent medium support selective Ln uptake.

The release of organic acids facilitates heterotrophic bioleaching (68, 69). Polyhydroxyl carboxylic acids such as D-glucono-1,5-lactone can chelate metals, including Ln (70, 71). One of the most strongly upregulated genes in response to Nd and the Ln cocktail encodes a FAD-dependent glucose 1-dehydrogenase (RHAL1_01212), which catalyzes the reaction $\beta$-D-glucose + NAD(P)$^+$ $\leftrightarrow$ D-glucono-1,5-lactone + NAD(P)H + H$^+$. We noticed this gene before when strain RH AL1 was grown with pyruvate in presence and absence of La (30). We have no robust data if D-glucono-1,5-lactone is secreted and functions as a chelator in strain RH AL1, but our data warrant further investigations on organic acids being used by strain RH AL1 in the context of Ln chelation and uptake. Chelating heavier Ln can reduce the uptake of non-utilizable Ln, for instance, by blocking porins.

High intracellular concentrations of (non-)utilizable metals cause toxicity and must be avoided by homeostatic mechanisms. In the case of Nd and Ln cocktail incubations, the simultaneous upregulation of heavy metal efflux mechanisms and downregulation of Fe uptake systems suggest accidental Ln uptake through the latter. The simultaneous release of a chelator and downregulation of the machinery needed for the uptake of metal-chelator complexes was described as a strategy to deal with elevated copper levels in *Pseudomonas aeruginosa* (72). The downregulation of *ebbB* and *ebbD*, coding for part of the machinery needed for transmitting energy between cytoplasmic and outer membrane (73), in RH AL1 in response to Nd and the Ln cocktail indicates that the energy coupling needed to drive TonB-dependent transport is reduced.

Different La concentration and Ln elements triggered the differential expression of genes of the *ssuABCDE* operon. SsuABCDE are responsible for taking up and oxidizing alkane sulfonates. Alkanesulfonates were not present in the used medium, and Beijerinckiaceae bacterium RH AL1 is not known to produce them. In *Acinetobacter oleivorans* DR1 and several other bacteria, including *Acinetobacter baumannii*, *Pseudomonas aeruginosa* PAO1, *Pseudomonas alloputida* KT2440, *Corynebacterium glutamicum*, and *Escherichia coli K-12*, *ssuABCDE* were shown to be upregulated in response to oxidative stress (74). Fighting oxidative stress requires increased amounts of sulfur as oxidative stress-sensing proteins and detoxifying enzymes are characterized by Fe-S clusters and disulfide bonds (75, 76). Our RNAseq data could indicate that different Ln elements and La concentrations cause different degrees of oxidative stress in Beijerinckiaceae bacterium RH AL1.

## Concluding remarks

We showed that different Ln elements affect many genes, tied to various pathways in Beijerinckiaceae bacterium RH AL1. These included aspects assumed to be regulated by Ca, and we postulate that Ln interfere with or complement the physiological role of Ca. Our findings suggest that strain RH AL1 can distinguish between different (utilizable) Ln. Not all of our findings do implicate causality, but they support the possibility that Ln play a diverse role in bacterial physiology. Understanding this role will facilitate tuning Ln-dependent metabolism toward biotechnological applications and a more sustainable use of these key resources of the 21$^{st}$ century.

## MATERIALS AND METHODS

### Cultivation

Beijerinckiaceae bacterium RH AL1 was grown using MM2 medium (77). Incubations were done at room temperature while shaking (110 rpm). Pre-cultures were grown

on sodium pyruvate (0.2%, wt/vol; 18.175 mM) as the carbon source and without Ln in acid-washed 200-mL serum bottles sealed with boiled and sterilized butyl rubber stoppers. Biomass was collected by centrifugation and repeatedly washed before being used as inoculum. Cultivation experiments were performed in triplicates ($n = 3$) in acid-washed 150-mL Erlenmeyer flasks with cellulose stoppers and methanol (0.5%, vol/vol; 123 mM) as the carbon source. Cultures were supplemented with either different concentrations of La (50 nM, 1 µM) or different lanthanide elements [La, Nd; 1 µM, lanthanide cocktail (Ce, Nd, Dy, Ho, Er, Yb)]. Ln concentrations were chosen based on previous cultivation experiments (23), with 50 nM representing the minimum Ln concentration required for growth at laboratory conditions and 1 µM Ln representing the optimum. The Ln cocktail was prepared by equimolarly pooling 1 mM solutions of the respective elements. The 1 mM solutions were prepared from lanthanide trichloride salts (Sigma-Aldrich, Taufkirchen, Germany). The concentration of the different elements in MM2 medium after adding the Ln cocktail (target concentration 1 µM Ln) was determined by ICP-MS (8900 Triple Quadrupole ICP-MS, Agilent Technologies, Waldbronn, Germany). The Ln concentration in the MM2 medium supplemented with the Ln cocktail was 0.89 µM Ln; concentrations ranged from 0.12 and 0.17 µM for the different Ln elements (Table S2). Growth was monitored by spectrophotometry ($\lambda = 600$ nM) using a Synergy H4 Hybrid reader (Agilent Technologies). Cultures were grown to the stationary phase, and samples for downstream, RNAseq-based transcriptome analysis were taken during the mid- to late-exponential phase ($t_1$). Additional samples were taken for cell counts based on quantitative PCR (qPCR) ($t_0$, beginning of incubation; $t_2$, end of incubation). Incubations with the Ln cocktail were repeated ($n = 3$) to collect cell-free supernatant samples during late-exponential phase for subsequent elemental analysis via ICP-MS to check for Ln depletion during incubation (Table S2).

## DNA extraction and qPCR

Genomic DNA was extracted following standard protocols using the NucleoSpin Microbial DNA Mini kit (Macherey-Nagel, Düren, Germany). DNA was quantified using a Qubit 3 fluorometer in combination with dsDNA BR or HS assay kits (Thermo Fisher Scientific, Frankfurt, Germany). DNA integrity was checked through spectrophotometry and DNA agarose gel electrophoresis. Details concerning qPCR can be found in the supplementary material.

## RNA extraction, mRNA enrichment, and sequencing library preparation

Total RNA was extracted and mRNA enriched from biomass samples ($n = 3$) collected in the mid- to late- exponential growth phase based on previously described methods (30, 78). Sequencing libraries were prepared with the NEBNext Ultra II Directional RNA library prep kit for Illumina (New England Biolabs, Ipswich, Massachusetts, USA) and checked by chip-based, high-resolution gel electrophoresis with a Bioanalyzer instrument and DNA 7500 Pico reagents (Agilent Technologies).

## Sequencing and data pre-processing

An equimolar pool of libraries was subjected to Illumina sequencing ($2 \times 100$ bp, paired ends) with a NovaSeq 6000 instrument and an SP flowcell (Illumina, San Diego, California, USA). Sequencing was carried out by the sequencing core facility of the Leibniz Institute on Aging-Fritz Lipmann Institute. The quality of raw and later trimmed sequences was checked with *FastQC* (v0.11.9) (79). Data pre-processing was done as described previously (30) and is outlined in the supplementary material.

## Differential gene expression analysis

Differential gene expression analysis was done using the packages *edgeR* (v3.20.9) (80), *limma* (v3.50.0) (81), *mixOmics* (v6.18.1) (82), *HTSFilter* (v1.34.0) (83), and *bigPint* (v1.10.0)

(84), in the R software framework for statistical computing (v3.5.1) (85). Further details are given in the supplementary material. Differentially expressed genes were filtered based on the $\log_2$FC, false discovery rate-corrected $P$ value (Fig. S4) and absolute gene expression in $\log_2$CPM. Genes with changes in gene expression above |0.58| $\log_2$FC, gene expression values higher than 4 ($\log_2$CPM), and $P$-values smaller than 0.05 were considered for downstream analysis.

## Overrepresentation analysis

Subsets of differentially expressed genes were subjected to overrepresentation analysis using the functions *enrichKEGG* and *enrichMKEGG* ($P$-value cutoff 0.05, $P$-value adjustment Bejamini-Hochberg procedure) from the R package *clusterProfiler* (v. 4.4.3) (86, 87) and the available genome annotation from the KEGG GENOMES databases (identifiers: T06029, "bbar").

## Motility assay

To investigate the motility of Beijerinckiaceae bacterium strain RH AL1 when grown with different Ln elements and concentrations (La, Nd, Ln cocktail; 5 nM, 10 nM, 50 nM, 100 nM, 1 µM, 5 µM, 10 µM), incubations were done with MM2 medium (77), semi-solidified with agar (0.4%, wt/vol) and contained 2,3,5-triphenyltetrazolium chloride (0.005%, wt/vol) (Carl Roth GmbH & Co. Kg, Karlsruhe, Germany) (88). Incubations were done in six-well plates with methanol (0.5%, vol/vol, 123 mM) as carbon and energy source. A preculture grown to stationary phase with pyruvate (0.2%, wt/vol, 18.175 mM) as a carbon source and in absence of lanthanides was washed five times with 10-mL basal MM2 medium and used as inoculum for the motility assay. The evaluation was performed 14 days after inoculation. Motility-assay data were tested for normal distribution (Kolmogorov-Smirnov test), outliers (Dean-Dixon test), and trend (Neumann trend test). Statistical analysis was performed with Microsoft Excel based on a 95% confidence interval.

## Electron microscopy

Sample preparation for TEM, FFTEM, and EDX analyses was carried out according to reference (30). Sample preparation is outlined in the supplementary material. The deconvolution of EDX spectra was done using the Quantax software (Bruker, Berlin, Germany).

## Analysis of TEM pictures

Cell areas and the areas occupied by PHB vacuoles were determined using *ImageJ* (v. 1.52r) (89, 90) and its freehand selection tool. The analysis was done for three images (magnification, 4,000×; image area, 540.5 µm$^2$), and between 27 and 30 cells were analyzed per condition (La, Nd, cocktail). Differences between the conditions with respect to the cell area, the area occupied by PHB, and the proportion of the cell area made up by PHB were tested for significance by one-way ANOVA, combined with a post-hoc Tukey-Kramer test.

## Figure generation

Plotting was done with the R software framework (v. 4.2.1) (91) using the packages *ggplot2* (v. 3.3.6) (92), *gplots* (v. 3.1.3) (93), *cowplot* (v. 1.1.1) (94), and *upsetR* (v. 1.4.0) (95), including their respective dependencies. Figures were finalized with inkscape (https://inkscape.org/).

## ACKNOWLEDGMENTS

C.E.W. thanks N. Cecilia Martinez-Gomez, Nathan M. Good (UC Berkeley, CA, USA), and Lena Daumann (Heinrich Heine University Düsseldorf, Germany) for sharing and discussing unpublished data.

This research was funded by the Deutsche Forschungsgemeinschaft (DFG) through grant WE6579/4-1 and supported by SFB 1127 ChemBioSys, project number 239748522. We acknowledge additional support by the DFG (project number 512648189) and the Thueringer Universitaets- und Landesbibliothek Jena for open access publishing.

## AUTHOR AFFILIATIONS

[1]Institute of Biodiversity, Aquatic Geomicrobiology, Friedrich Schiller University, Jena, Germany

[2]Electron Microscopy Center, Jena University Hospital, Jena, Germany

## AUTHOR ORCIDs

Carl-Eric Wegner ⓘ http://orcid.org/0000-0002-4339-6602

## FUNDING

| Funder | Grant(s) | Author(s) |
| --- | --- | --- |
| Deutsche Forschungsgemeinschaft (DFG) | WE6579/4-1 | Carl-Eric Wegner |
| | | Linda Gorniak |
| Deutsche Forschungsgemeinschaft (DFG) | SFB 1127 ChemBioSys project number 239748522 | Carl-Eric Wegner |
| Deutsche Forschungsgemeinschaft (DFG) | open access oublishing project number 512648189 | Carl-Eric Wegner |

## AUTHOR CONTRIBUTIONS

Linda Gorniak, Conceptualization, Formal analysis, Investigation, Writing – original draft | Julia Bechwar, Formal analysis, Investigation, Writing – original draft | Martin Westermann, Formal analysis, Investigation, Methodology, Writing – original draft | Frank Steiniger, Formal analysis, Investigation, Methodology, Writing – original draft | Carl-Eric Wegner, Conceptualization, Formal analysis, Funding acquisition, Project administration, Supervision, Writing – original draft, Writing – review and editing

## DATA AVAILABILITY

RNA-seq data sets can be accessed via EBI/ENA ArrayExpress [accession: E-MTAB-12015]. The genome of Beijerinckiaceae bacterium RH AL1 is available via the EBI/ENA accession numbers: LR590083 and LR699074 (genome and plasmid). We also provide details about sequence data processing and differential gene expression analysis via the Open Science Framework (https://osf.io/) (https://osf.io/p2nf6/?view_only=b83c7bbd806b43bdac419ebc8117eaa0). A snakemake workflow for sequence data processing is available on github (https://github.com/wegnerce/smk_rna-seq).

## ADDITIONAL FILES

The following material is available online.

### Supplemental Material

**Supplementary Material (Spectrum00867-23-S0001.pdf).** Additional experimental details, captions for supplementary figures (figures included) and tables.

**Supplementary Tables (Spectrum00867-23-S0002.xls).** All supplementary tables combined in one spreadsheet.

## Open Peer Review

**PEER REVIEW HISTORY (review-history.pdf).** An accounting of the reviewer comments and feedback.

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
