## [Reviewer comments · Microbiology Spectrum]

Microbiology Spectrum

Different lanthanide elements induce strong gene expression changes in a lanthanide-accumulating methylotroph

Linda Gorniak, Julia Bechwar, Frank Steiniger, Martin Westermann, and Carl-Eric Wegner

Corresponding Author(s): Carl-Eric Wegner, Friedrich-Schiller-Universitat Jena

Review Timeline:

Submission Date:	March 7, 2023
Editorial Decision:	July 11, 2023
Revision Received:	August 28, 2023
Editorial Decision:	September 22, 2023
Revision Received:	September 22, 2023
Accepted:	September 25, 2023

Editor: Jannell Bazurto

Reviewer(s): Disclosure of reviewer identity is with reference to reviewer comments included in decision letter(s). The following individuals involved in review of your submission have agreed to reveal their identity: Eric Lee Bruger (Reviewer #1); Srujana Samhita S Yadavalli (Reviewer #2)

Transaction Report:

DOI: <https://doi.org/10.1128/spectrum.00867-23>

July 11, 2023

Dr. Carl-Eric Wegner
Friedrich-Schiller-Universität Jena
Institute of Biodiversity, Aquatic Geomicrobiology
Dornburger Str. 159
Jena 07743
Germany

Re: Spectrum00867-23 (Different lanthanide elements induce strong gene expression changes in a lanthanide-accumulating methylotroph)

Dear Dr. Carl-Eric Wegner:

Many thanks for submitting your manuscript entitled "Different lanthanide elements induce strong gene expression changes in a lanthanide-accumulating methylotroph" for consideration by Microbiology Spectrum. The expertise of two external referees was requested and as you will see, both referees were positive regarding your manuscript, but raised a number of issues that can be tackled at the textual and data visualization levels. In particular, Reviewer #1 noted significant issues regarding clarity of communication and the lack of data to support several mechanistic claims. In general, we do believe that tackling all of the Referees' criticisms will result in amendments that will further strengthen the validity paper.

Link Not Available

Sincerely,

Jannell Bazurto

Journals Department
Reviewer comments:

Reviewer #1 (Comments for the Author):

Microbiology Spectrum Review, June 2023

Article Title: Different lanthanide elements induce strong gene expression changes in a lanthanide-accumulating methylotroph
Article Authors: Linda Gorniak , Julia Bechwar, Martin Westermann, Frank Steiniger, Carl-Eric Wegner

Summary

The manuscript documents the differential impacts of alternative lanthanide rare earth metals upon gene expression within the recently characterized methylotrophic bacterium Beijerinckiaceae bacterium RH AL1. Lanthanides have recently garnered much interest as an alternative metal enabling methylotrophic metabolism, primarily by serving as a cofactor for XoxF-like dehydrogenase enzymes. The authors observe strong changes in gene expression depending on what lanthanides cells are grown in the presence of, including diverse metabolic processes such as chemotaxis and PHA synthesis. The study is valuable to the growing field of lanthanide-focused biology, as it takes a more in-depth view of the impacts of alternative and single vs. combinatorial lanthanides upon gene expression. In particular, comparisons of how different (utilizable) lanthanides affect metabolism have rarely been done, and strain RH AL1 appears to distinguish between different lanthanide elements.

Major comments

1. The manuscript needs significant rewriting to make it clearer and more impactful. As an example, text descriptions of the groups being compared are confusing at times. The authors should also preface sections in the discussion section. They often dive right into paragraphs with detailed statements without providing the appropriate context, rationale, and bigger picture themes, and need to set these up more carefully. For example, in viewing the paragraph focused on calcium (Lines 289-296), the first sentence is not required, and the paragraph would read better if it started from the second sentence. This section and others ask the reader to connect too many dots rather than stating them more explicitly (or testing them directly in the study). Other paragraphs are rather unfocused: e.g., Lines 348-353 read as just a sequence of facts about sulfur metabolism rather than a cohesive discussion of the potential impacts that lanthanides have upon sulfur metabolism.
2. The authors overstate the impact of the findings at times, such as suggesting the presence of different uptake mechanisms for different lanthanides in the abstract, something that is not convincingly broached at all with the experiments performed in this study. On Lines 337-341, the authors mention the upregulation of "heavy metal efflux mechanisms" (should be "genes"). This is a trend of expression and sets up a testable hypothesis over the specificity of these chelators, but is not conclusive. There are a few other places where the term "mechanism" is invoked but does not seem to be done so accurately.
3. The text has a lot of soft, "maybe so", kind of language, most noticeably in the discussion section, which leaves it unclear to the reader how seriously to take some of the statements being made.
4. Figures look nice but are not always referred to correctly/sufficiently. This includes both references in the text as well as the level of description provided in the accompanying figure legends.

Methods

1. Why do the authors start growth on pyruvate? Are lanthanides required for growth under this condition? This also constitutes a switch (pyruvate- to methanol-based growth) to methylotrophy, which could include related transcriptomic changes that will get lost in this analysis.
2. The setup for the transcriptomics is a bit confusing (and is why I suggest including an accompanying figure below). It seems that for this, all treatments should either be a) pregrown in the same lanthanide condition, or b) pregrown in a condition identical to that which it will be transferred into. If neither of these conditions is satisfied, then it feels like the cross-comparison between differently treated samples is problematic.
3. It would be informative for the authors to speak to the concentrations of lanthanides used in the study, as compared to measured or expected concentrations available in typical environmental conditions.

Figures

General:

1. Overall: Great in appearance. However, the manuscript needs more comprehensive descriptions in the figure legends.
2. It would be helpful to implement a schematic figure explaining how the experiments were set up and carried out for growth and transcriptomics, because the written explanations often don't completely clarify the details of it.

Specific:

-Figure 1: a) It would be much more accurate to target a timeframe within the exponential phase to get maximum td values. b)

Why are the doubling times in La rather different between the left and right panels? c) The legend is highly insufficient and doesn't include 1B. d) The redundancy in 1uM La in both panels is confusing. Where did the 1uM La data come from? The cultures from the left panel, or right?

-Figure 2: This figure had a number of aspects about it that made it difficult to follow. From a statistical standpoint, for comparisons like this it makes more sense to have more stringent cutoffs for statistical significance to be assessed. A more common FDR value to use as a significance cutoff is $FDR < 0.01$. In the figure legend, I think it makes better sense to have A-C labels at the beginning rather than the ends of statements. For panels B and C, it would be clearer to represent the comparisons with word descriptions rather than numbering 1-4. In Figure 2C, it does not make sense to have single comparisons rather than intersections between comparisons. In the case the single comparisons (2 and 3) are represented, why are the unique genes to 1 and 4 also not shown?

-Figure 3: I am unsure that this is accomplishing what it is seeking to. I would potentially separate the central cell illustration from the surrounding data panels into separate figures. Alternatively, it might make sense to provide more color-coding to the central illustrations to have processes align with the figure legend on the side.

-Figure 4 is not referred to directly anywhere in the text. Please adjust and insert appropriate reference points.

-Figure 5 refers to PHB, but the text does not explain that PHB is a specific type of PHA molecule until later in the discussion section.

Minor comments

General:

1. Report methanol and pyruvate concentrations as molar values, for better relevance and consistency with the lanthanide concentrations.
2. At least once, report the concentration of each single lanthanide compound within the combined cocktail treatment.
3. The authors refer to their focal strain in different ways (e.g., "Beijerinckiaceae bacterium RH AL1", "RH AL1", "strain AL1", etc.) throughout the paper. After initial introduction, choose one in order to maintain consistency throughout.
4. In some of the results, the authors report the trend of a comparison, but not the quantitative measure. I would suggest reporting \log_2 fold changes throughout for consistency and added information to the reader.
5. Unless the authors plan to refer to terms throughout, it might be confusing to define new terms like the lanthanome.

Specific:

- Line 28-29 - This sentence is vaguely worded, rephrase.
- Line 30 - Emphasizing the unknown or hypothetical genes feels like something to bring up later, but not in the abstract.
- Line 58 - The authors should define methylotrophy upon first usage.
- Line 113-114 - different time (t) designations are confusing
- Line 121 - What are the associated ODs with this? Can it really be considered mid-exponential phase?
- Line 122 - Confused about the cutoff of $|0.58|$, it feels a bit arbitrary. Is this explained anywhere?
- Lines 163-166 - These two sentences feel a bit uninformative and out-of-place for a results section. Are they really necessary? I would consider rephrasing or even cutting these statements from the manuscript.
- Line 196 - Add " \log_2FC " - also check throughout to make sure of its presence.
- Line 198-199 - Wording makes it unclear which one of the two is lower.
- Line 225 - Say more about the nature of the type of t-test performed.
- Line 212 - Why is only one number reported for RHAL1_02672 when 2 numbers are reported for the other genes listed here?
- Line 219 - Replace the "+" sign here.
- Line 233 - Replace the "+" sign here.
- Line 234 - How do you attribute distinct signals to specific lanthanides here? It is confusing.
- Line 236 - Use elemental abbreviations to be consistent throughout.
- Line 237 - Confused about why P is invoked here, there is no verbiage to preface why.
- Line 240-241 - Confusing way to start discussion section...
- Line 251 - Suggested change to start the sentence: "Based on these data..."
- Line 254 - A supplemental table of lanthanide names, since you are using so many of them, would be helpful to someone who does not regularly work with all these compounds.
- Line 257 - Insert text "in this study" in the sentence to reorient reader.

- Line 261 - do not define metalation anywhere in the text. A little hint to the reader would be helpful.
- Line 274 - "in a non-methylotrophic background" alone is quite vague - could you use text to expand more on this a bit?
- Line 283-284 - Is this sentence important/necessary?
- Line 286 - Is there something particularly interesting about pyruvate that makes it interesting/worth reporting on? Similarly, is the whole long following section focused upon Calcium completely necessary? The way the discussion is written, going back and forth between Ca and Ln is actually more confusing rather than informative.
- Line 297 - Typo.
- Line 300 - What type of sensitivity are you talking about?
- Line 313 - First mention of the term lanthanide phosphates. Should be broached earlier when P is first mentioned.
- Line 348 - Is sulfur metabolism being invoked or not really? Seems like really soft language considering a lot of space in text and figures elsewhere have already been devoted to sulfur.
- Line 355 - This is not necessary to be quoted or cited, in my opinion.
- Line 376 - Unclear if all samples were acclimated on 1) homologous conditions into which they were diluted, 2) if all were acclimated in identical condition, or 3) some hybrid of 1) and 2).
- Line 441 - Suggested rephrasing: "using the freehand selection tool in ImageJ..."
- Line 718-719 - Only 3 groups are listed here, list all 4 groups.
- Line 719 - Add a period.
- Line 719 - this might be the only place in the paper where the lanthanome is directly defined. It should be defined in the text as well.

Reviewer #2 (Comments for the Author):

This paper by Gorniak et al., investigates how different lanthanides affect gene expression and metabolism using lanthanide-dependent, and lanthanide-accumulating methylotroph, *Beijerinckiaceae* bacterium RH AL1. Lanthanide accumulation in RH AL1 was previously shown by this group and in this work the authors looked at different lanthanides and concentrations of lanthanum, to identify the differentially expressed genes. Overall the work is exciting and the conclusions are valid. It is quite interesting to see the differences and commonalities in response to different Ln elements. The effects of differential gene expression and their physiological significance are yet to be understood.

I have minor comments to further improve this manuscript.

1. Please include full forms of all abbreviations and details to the figure legends to help the reader understand the figures presented without referring back to the text in the results section. For instance in Fig. 1 - please mention La, Nd, cocktail stand for, panel A to the left shows lanthanum concentrations? Panel B is missing from the legend. In Panel B, what do copies refer to? Also, mention the number of replicates for the growth assays and experiments.
2. Please mention the number of replicates performed for the RNA-seq experiment for the 4 different conditions (preferably in the main text and figure legend)
3. What do the terms "overrepresentation" and "gene enrichment" mean in terms of biology? Are all the genes that are identified from the gene enrichment analysis also identified as overrepresented in the overrepresentation analysis?
4. It is interesting to see hits in folate biosynthesis, secondary metabolism, two-component systems, etc but the gene expression appears different under La versus cocktail or pairwise comparisons. The authors could elaborate/discuss a bit more about the potential physiological implications of these changes in different pathways with respect to separate conditions in the discussion.

Staff Comments:

Preparing Revision Guidelines

For complete guidelines on revision requirements, please see the journal Submission and Review Process requirements at

<https://journals.asm.org/journal/Spectrum/submission-review-process>. **Submissions of a paper that does not conform to Microbiology Spectrum guidelines will delay acceptance of your manuscript. "**

Please return the manuscript within 60 days; if you cannot complete the modification within this time period, please contact me. If you do not wish to modify the manuscript and prefer to submit it to another journal, please notify me of your decision immediately so that the manuscript may be formally withdrawn from consideration by Microbiology Spectrum.

Microbiology Spectrum Review, June 2023

Article Title: Different lanthanide elements induce strong gene expression changes in a lanthanide-accumulating methylotroph

Article Authors: Linda Gorniak , Julia Bechwar, Martin Westermann, Frank Steiniger, Carl-Eric Wegner

Summary

The manuscript documents the differential impacts of alternative lanthanide rare earth metals upon gene expression within the recently characterized methylotrophic bacterium *Beijerinckiaceae* bacterium RH AL1. Lanthanides have recently garnered much interest as an alternative metal enabling methylotrophic metabolism, primarily by serving as a cofactor for XoxF-like dehydrogenase enzymes. The authors observe strong changes in gene expression depending on what lanthanides cells are grown in the presence of, including diverse metabolic processes such as chemotaxis and PHA synthesis. The study is valuable to the growing field of lanthanide-focused biology, as it takes a more in-depth view of the impacts of alternative and single vs. combinatorial lanthanides upon gene expression. In particular, comparisons of how different (utilizable) lanthanides affect metabolism have rarely been done, and strain RH AL1 appears to distinguish between different lanthanide elements.

Major comments

1. The manuscript needs significant rewriting to make it clearer and more impactful. As an example, text descriptions of the groups being compared are confusing at times. The authors should also preface sections in the discussion section. They often dive right into paragraphs with detailed statements without providing the appropriate context, rationale, and bigger picture themes, and need to set these up more carefully. For example, in viewing the paragraph focused on calcium (Lines 289-296), the first sentence is not required, and the paragraph would read better if it started from the second sentence. This section and others ask the reader to connect too many dots rather than stating them more explicitly (or testing them directly in the study). Other paragraphs are rather unfocused: e.g., Lines 348-353 read as just a sequence of facts about sulfur metabolism rather than a cohesive discussion of the potential impacts that lanthanides have upon sulfur metabolism.
2. The authors overstate the impact of the findings at times, such as suggesting the presence of different uptake mechanisms for different lanthanides in the abstract, something that is not convincingly broached at all with the experiments performed in this study. On Lines 337-341, the authors mention the upregulation of “heavy metal efflux mechanisms” (should be “genes”). This is a trend of expression and sets up a testable hypothesis over the specificity of these chelators, but is not conclusive. There are a few other places where the term “mechanism” is invoked but does not seem to be done so accurately.
3. The text has a lot of soft, “maybe so”, kind of language, most noticeably in the discussion section, which leaves it unclear to the reader how seriously to take some of the statements being made.
4. Figures look nice but are not always referred to correctly/sufficiently. This includes both references in the text as well as the level of description provided in the accompanying figure legends.

Methods

1. Why do the authors start growth on pyruvate? Are lanthanides required for growth under this condition? This also constitutes a switch (pyruvate- to methanol-based growth) to methylotrophy, which could include related transcriptomic changes that will get lost in this analysis.
2. The setup for the transcriptomics is a bit confusing (and is why I suggest including an accompanying figure below). It seems that for this, all treatments should either be a) pregrown in the same lanthanide condition, or b) pregrown in a condition identical to that which it will be transferred into. If neither of these conditions is satisfied, then it feels like the cross-comparison between differently treated samples is problematic.
3. It would be informative for the authors to speak to the concentrations of lanthanides used in the study, as compared to measured or expected concentrations available in typical environmental conditions.

Figures

General:

1. Overall: Great in appearance. However, the manuscript needs more comprehensive descriptions in the figure legends.
2. It would be helpful to implement a schematic figure explaining how the experiments were set up and carried out for growth and transcriptomics, because the written explanations often don't completely clarify the details of it.

Specific:

-Figure 1: a) It would be much more accurate to target a timeframe within the exponential phase to get maximum td values. b) Why are the doubling times in La rather different between the left and right panels? c) The legend is highly insufficient and doesn't include 1B. d) The redundancy in 1uM La in both panels is confusing. Where did the 1uM La data come from? The cultures from the left panel, or right?

-Figure 2: This figure had a number of aspects about it that made it difficult to follow. From a statistical standpoint, for comparisons like this it makes more sense to have more stringent cutoffs for statistical significance to be assessed. A more common FDR value to use as a significance cutoff is $FDR < 0.01$. In the figure legend, I think it makes better sense to have A-C labels at the beginning rather than the ends of statements. For panels B and C, it would be clearer to represent the comparisons with word descriptions rather than numbering 1-4. In Figure 2C, it does not make sense to have single comparisons rather than intersections between comparisons. In the case the single comparisons (2 and 3) are represented, why are the unique genes to 1 and 4 also not shown?

-**Figure 3:** I am unsure that this is accomplishing what it is seeking to. I would potentially separate the central cell illustration from the surrounding data panels into separate figures. Alternatively, it might make sense to provide more color-coding to the central illustrations to have processes align with the figure legend on the side.

-**Figure 4** is not referred to directly anywhere in the text. Please adjust and insert appropriate reference points.

-**Figure 5** refers to PHB, but the text does not explain that PHB is a specific type of PHA molecule until later in the discussion section.

Minor comments

General:

1. Report methanol and pyruvate concentrations as molar values, for better relevance and consistency with the lanthanide concentrations.
2. At least once, report the concentration of each single lanthanide compound within the combined cocktail treatment.
3. The authors refer to their focal strain in different ways (e.g., "*Beijerinckiaceae* bacterium RH AL1", "RH AL1", "strain AL1", etc.) throughout the paper. After initial introduction, choose one in order to maintain consistency throughout.
4. In some of the results, the authors report the trend of a comparison, but not the quantitative measure. I would suggest reporting log₂ fold changes throughout for consistency and added information to the reader.
5. Unless the authors plan to refer to terms throughout, it might be confusing to define new terms like the lanthanome.

Specific:

- Line 28-29 – This sentence is vaguely worded, rephrase.
- Line 30 – Emphasizing the unknown or hypothetical genes feels like something to bring up later, but not in the abstract.
- Line 58 – The authors should define methylotrophy upon first usage.
- Line 113-114 – different time (t) designations are confusing
- Line 121 – What are the associated ODs with this? Can it really be considered mid-exponential phase?
- Line 122 – Confused about the cutoff of |0.58|, it feels a bit arbitrary. Is this explained anywhere?

- Lines 163-166 – These two sentences feel a bit uninformative and out-of-place for a results section. Are they really necessary? I would consider rephrasing or even cutting these statements from the manuscript.
- Line 196 – Add “log₂FC” - also check throughout to make sure of its presence.
- Line 198-199 – Wording makes it unclear which one of the two is lower.
- Line 225 – Say more about the nature of the type of t-test performed.
- Line 212 – Why is only one number reported for RHAL1_02672 when 2 numbers are reported for the other genes listed here?
- Line 219 – Replace the “+” sign here.
- Line 233 – Replace the “+” sign here.
- Line 234 – How do you attribute distinct signals to specific lanthanides here? It is confusing.
- Line 236 – Use elemental abbreviations to be consistent throughout.
- Line 237 – Confused about why P is invoked here, there is no verbiage to preface why.
- Line 240-241 – Confusing way to start discussion section...
- Line 251 – Suggested change to start the sentence: “Based on these data...”
- Line 254 – A supplemental table of lanthanide names, since you are using so many of them, would be helpful to someone who does not regularly work with all these compounds.
- Line 257 – Insert text “in this study” in the sentence to reorient reader.
- Line 261 – do not define metalation anywhere in the text. A little hint to the reader would be helpful.
- Line 274 – “in a non-methylotrophic background” alone is quite vague - could you use text to expand more on this a bit?
- Line 283-284 – Is this sentence important/necessary?
- Line 286 – Is there something particularly interesting about pyruvate that makes it interesting/worth reporting on? Similarly, is the whole long following section focused upon Calcium completely necessary? The way the discussion is written, going back and forth between Ca and Ln is actually more confusing rather than informative.
- Line 297 – Typo.
- Line 300 – What type of sensitivity are you talking about?
- Line 313 – First mention of the term lanthanide phosphates. Should be broached earlier when P is first mentioned.
- Line 348 – Is sulfur metabolism being invoked or not really? Seems like really soft language considering a lot of space in text and figures elsewhere have already been devoted to sulfur.
- Line 355 – This is not necessary to be quoted or cited, in my opinion.
- Line 376 – Unclear if all samples were acclimated on 1) homologous conditions into which they were diluted, 2) if all were acclimated in identical condition, or 3) some hybrid of 1) and 2).
- Line 441 – Suggested rephrasing: “using the freehand selection tool in ImageJ...”
- Line 718-719 – Only 3 groups are listed here, list all 4 groups.
- Line 719 – Add a period.
- Line 719 – this might be the only place in the paper where the lanthanome is directly defined. It should be defined in the text as well.

General remarks:

Additional data: Carried electron microscopy combined with elemental analysis by means of energy-dispersive X-ray spectroscopy suggested that different lanthanide (Ln) elements are preferentially taken up when *Beijerinckiaceae* bacterium RH AL1 is cultivated with a cocktail of light and heavy lanthanides. We repeated the incubations to strengthen this point. We took samples during the late-exponential growth phase and subjected cell-free supernatant samples to elemental analysis through ICP-MS (inductively coupled plasma mass spectrometry). Obtained results revealed that the supernatant samples were depleted of lighter lanthanides (Ce, Nd), while heavier lanthanides (Dy, Ho, Er, Yb) were still detectable. We detected $3.02 \pm 1.58\%$ of the initially present Er, and $75.93 \pm 3.58\%$ of the initially present Yb. These findings support our claim that *Beijerinckiaceae* bacterium RH AL1 selectively takes up lanthanides, and we added them to the manuscript, **see lines 285-290**, and **Table S16**.

The lack of La in intracellular inclusions, when strain RH AL1 was cultivated with our Ln cocktail, was puzzling and prompted us to determine the Ln content of our medium after adding the Ln cocktail. ICP-MS revealed that the Ln cocktail contained Ce, Nd, Dy, Ho, Er, Yb, but that we erroneously stated that it contains La. We corrected this in the main text, see lines e.g. **124-125**, describe the preparation of the cocktail in the methods section, **see lines 461-470**, and added the elemental analysis data, see **Table S2**.

Additional methods: We added details regarding the carried out ICP-MS analysis to the methods section, see **lines 465-467**. Regarding RNAseq data processing, we prepared a reproducible data processing workflow, which we reference in the methods part, see **lines 571-572**, and which is available online (https://github.com/wegnerce/smk_rnaseq).

Responses to reviewer comments:

Reviewer #1:

Major comments:

*The manuscript documents the differential impacts of alternative lanthanide rare earth metals upon gene expression within the recently characterized methylotrophic bacterium *Beijerinckiaceae* bacterium RH AL1. Lanthanides have recently garnered much interest as an alternative metal enabling methylotrophic metabolism, primarily by serving as a cofactor for XoxF-like dehydrogenase enzymes. The authors observe strong changes in gene expression depending on what lanthanides cells are grown in the presence of, including diverse metabolic processes such as chemotaxis and PHA synthesis. The study is valuable to the growing field of lanthanide-focused biology, as it takes a more in-depth view of the impacts of alternative and single vs. combinatorial lanthanides upon gene expression. In particular, comparisons of how different (utilizable) lanthanides affect metabolism have rarely been done, and strain RH AL1 appears to distinguish between different lanthanide elements.*

Thank you for your assessment of the manuscript. You can find our answers to your comments below.

1. The manuscript needs significant rewriting to make it clearer and more impactful. As an example, text descriptions of the groups being compared are confusing at times. The authors should also preface sections in the discussion section. They often dive right into paragraphs with detailed statements without providing the appropriate context, rationale, and bigger picture themes, and need to set these up more carefully. For example, in viewing the paragraph focused on calcium (Lines 289-296), the first sentence is not required, and the paragraph would read better if it started from the second sentence. This section and others ask the reader to connect too many dots rather than stating them more explicitly (or testing them directly in the study). Other paragraphs are rather unfocused: e.g., Lines 348-353 read as just a sequence of facts about sulfur metabolism rather than a cohesive discussion of the potential impacts that lanthanides have upon sulfur metabolism.

We agree that the manuscript lacked in parts clarity and readability, during the revisions we worked on improving both. Please see for instance the reworked sections dedicated to Ca and Ln (**lines 350-380**) and alkanesulfonates and Ln (**lines 423-434**) in the discussion, or the partially reworked introduction (**lines 102-107**).

2. The authors overstate the impact of the findings at times, such as suggesting the presence of different uptake mechanisms for different lanthanides in the abstract, something that is not convincingly broached at all with the experiments performed in this study. On Lines 337-341, the authors mention the upregulation of "heavy metal efflux mechanisms" (should be "genes"). This is a trend of expression and sets up a testable hypothesis over the specificity of these chelators, but is not conclusive. There are a few other places where the term "mechanism" is invoked but does not seem to be done so accurately.

We have weakened our statements referring to mechanisms, **see for instance lines 388-398**, where we no longer state that different Ln elements are deposited by different mechanisms.

3. The text has a lot of soft, "maybe so", kind of language, most noticeably in the discussion section, which leaves it unclear to the reader how seriously to take some of the statements being made.

You are absolutely right and we would love to have more definitive answers. We modified the text to make it less fuzzy, e.g. the sections about Ca and alkanesulfonates (**lines 350-380 + lines 423-434**). Our findings raise a lot of questions that we can not immediately answer, but that we are striving to address in follow-up work. One focus is, for instance, intracellular Ln storage and the potential link to PHA metabolism. Working with non-model microorganisms is challenging as a lot of resources and techniques are not easily available. Having, for instance, genetic tools would facilitate testing hypotheses in a (more) straightforward way. So far we lack them, but establishing them is one of our main interests right now.

4. Figures look nice but are not always referred to correctly/sufficiently. This includes both references in the text as well as the level of description provided in the accompanying figure legends.

We worked on the clarity of the figure legends (see for instance the modified legend of **Figure 2**) and tried to incorporate figures better in the main text (see for instance **lines 298-301 + 350**).

Methods:

1. Why do the authors start growth on pyruvate? Are lanthanides required for growth under this condition? This also constitutes a switch (pyruvate- to methanol-based growth) to methylotrophy, which could include related transcriptomic changes that will get lost in this analysis.

We previously reported that Beijerinckiaceae bacterium RH AL1 accumulates lanthanides intracellularly (Wegner et al., 2021; <https://doi.org/10.1128/AEM.03144-20>) during lanthanide-dependent growth with methanol as the carbon source. Pyruvate was used as the carbon source for the pre-cultures to prevent the carryover of lanthanides, which could impact gene expression. If pre-cultures were grown with methanol and La, our *de facto* standard conditions, and Nd or the Ln cocktail are added for subsequent incubations, residual La from the inoculum could mask effects caused by Nd or the Ln cocktail.

2. The setup for the transcriptomics is a bit confusing (and is why I suggest including an accompanying figure below). It seems that for this, all treatments should either be a) pregrown in the same lanthanide condition, or b) pregrown in a condition identical to that which it will be transferred into. If neither of these conditions is satisfied, then it feels like the cross-comparison between differently treated samples is problematic.

We have added a figure to the manuscript that explains the cultivation setup, **see Figure 1**. As outlined above, pyruvate was chosen as the carbon source for pre-cultures to prevent lanthanide carryover. During the incubations for downstream RNAseq analysis, all cultures were grown with methanol as the carbon source. The cultures only differed with respect to Ln supplementation (different La concentration, and different Ln elements added).

3. It would be informative for the authors to speak to the concentrations of lanthanides used in the study, as compared to measured or expected concentrations available in typical environmental conditions.

This is a good point, we added information about environmental concentrations of lanthanides to the main text, **see lines 310-315**. Please note that it is not easy to link concentrations in the environment (usually given in ppm when dealing with terrestrial systems) with concentrations used in the lab.

Figures:

General:

1. Overall: Great in appearance. However, the manuscript needs more comprehensive descriptions in the figure legends.

The figure legends have been revised to increase readability, see for instance the rewritten legend of **Figure 3**.

2. It would be helpful to implement a schematic figure explaining how the experiments were set up and carried out for growth and transcriptomics, because the written explanations often don't completely clarify the details of it.

We have added a figure to the manuscript that explains the cultivation setup, **see Figure 1**.

Specific:

Figure 1: a) It would be much more accurate to target a timeframe within the exponential phase to get maximum t_d values. b) Why are the doubling times in La rather different between the left and right panels? c) The legend is highly insufficient and doesn't include 1B. d) The redundancy in 1 μ M La in both panels is confusing. Where did the 1 μ M La data come from? The cultures from the left panel, or right?

a) We have targeted a timeframe within the exponential phase for the calculation of μ and t_d . We used time intervals that span at least three measurements. We have revisited our calculations and determined t_d and μ values and corrected them. The growth curve data was added to the supplement, **see Table S3**. b) We have seen differences of around 10% regarding μ and t_d in the past between different incubation runs. c) The legend of the former **Figure 1**, now **Figure 2**, was rewritten to make it more understandable. d) The carried out RNAseq-based gene expression analyses were based on two sets of incubations, one with two different La concentrations (50 nM and 1 μ M), and one with different lanthanide supplementation. Both sets of incubations included cultures supplemented with 1 μ M La. We are not sure what you are referring to when you say 1 μ M La data. In **Figures 3-5**, the 1 μ M data (considered for comparison 1) was generated based on samples from the incubation shown in the left panel of now **Figure 2**.

Figure 2: This figure had a number of aspects about it that made it difficult to follow. From a statistical standpoint, for comparisons like this it makes more sense to have more stringent cutoffs for statistical significance to be assessed. A more common FDR value to use as a significance cutoff is $FDR < 0.01$. In the figure legend, I think it makes better sense to have A-C labels at the beginning rather than the ends of statements. For panels B and C, it would be clearer to represent the comparisons with word descriptions rather than numbering 1-4. In Figure 2C, it does not make sense to have single comparisons rather than intersections between comparisons. In the case the single comparisons (2 and 3) are represented, why are the unique genes to 1 and 4 also not shown?

We agree, that more stringent FDR cutoffs lead to more robust results. However, an FDR cutoff of < 0.05 is commonly used for RNAseq-based differential gene expression analysis (exemplary studies that look at differential gene expression analysis from a more technical view: <https://doi.org/10.1186/s13059-019-1716-1>; <https://doi.org/10.1002/cpmb.68>). One of the few RNAseq studies addressing Ln-dependent gene expression changes (<https://doi.org/10.1038/s41598-019-41043-1>) in *M. extorquens* AM1 used an FDR cutoff of 0.15. A-C labels are now placed at the beginning of statements throughout the manuscript. For panels B and C word descriptions have been added in addition to the numbering. Panel C only shows sets of genes > 100 , the number of unique genes for 1 and 4 was lower. We clarified this in the revised legend.

Figure 3: I am unsure that this is accomplishing what it is seeking to. I would potentially separate the central cell illustration from the surrounding data panels into separate figures. Alternatively, it might make sense to provide more color-coding to the central illustrations to have processes align with the figure legend on the side.

We agree, the cell illustration provided only little additional value. In the revised manuscript, the former **Figure 3** was split into **Figure 4** and **Figure 8**. **Figure 4** highlights changes in gene expression with respect to certain functional aspects using ridge plots, **Figure 8** uses the cell illustration to summarize our understanding of the impact of Ln on cellular physiology in Beijerinckiaceae bacterium RH AL1.

Figure 4 is not referred to directly anywhere in the text. Please adjust and insert appropriate reference points.

We now refer to this figure (**Figure 5**) in the main text, **see e.g. lines 221 and 238**.

Figure 5 refers to PHB, but the text does not explain that PHB is a specific type of PHA molecule until later in the discussion section.

PHB is now first defined in the main text in the context of gene expression changes linked to PHA metabolism, **see lines 204-206**.

Minor comments:

General:

1. Report methanol and pyruvate concentrations as molar values, for better relevance and consistency with the lanthanide concentrations.

We now report the concentrations as molar values, **see e.g. line 122 and lines 451 + 456**.

2. At least once, report the concentration of each single lanthanide compound within the combined cocktail treatment.

We added this information to the manuscript text (**lines 467-469**) and explain the preparation of the cocktail in the materials and methods section, **see lines 461-470**. The composition of the cocktail is also given in **Table S2**.

3. The authors refer to their focal strain in different ways (e.g., "Beijerinckiaceae bacterium RH AL1", "RH AL1", "strain AL1", etc.) throughout the paper. After initial introduction, choose one in order to maintain consistency throughout.

In the revised manuscript we consistently refer to Beijerinckiaceae bacterium RH AL1 or strain RH AL1.

4. In some of the results, the authors report the trend of a comparison, but not the quantitative measure. I would suggest reporting log₂ fold changes throughout for consistency and added information to the reader.

We checked that log₂FC values are given wherever we explicitly refer to gene expression changes.

5. Unless the authors plan to refer to terms throughout, it might be confusing to define new terms like the lanthanome.

The term "lanthanome" (<http://dx.doi.org/10.1021/jacs.8b12155>) is not really new anymore, it is meanwhile commonly used in literature. We define it upon the first usage in the revised manuscript, **see lines 218-224**.

Specific:

Line 28-29 - This sentence is vaguely worded, rephrase.

We omitted this sentence in the revised manuscript.

Line 30 - Emphasizing the unknown or hypothetical genes feels like something to bring up later, but not in the abstract.

In the revised version, we no longer mention unknown/hypothetical genes in the abstract.

Line 58 - The authors should define methylotrophy upon first usage.

Methylotrophy is now defined upon the first usage in the main text, **see lines 66-68**.

Line 113-114 - different time (t) designations are confusing

We agree, t_0 - t_2 are now defined in the main text and also in the legend of **Figure 2**, formerly **Figure 1**, **see lines 133-134 + 144**.

Line 121 - What are the associated ODs with this? Can it really be considered mid-exponential phase?

The corresponding OD values are now given, **see line 143**.

Line 122 - Confused about the cutoff of |0.58|, it feels a bit arbitrary. Is this explained anywhere?

A \log_2 FC of |0.58| is equivalent to a 50% increase/decrease in gene expression, we added this information to the main text, **see lines 144-145**.

Lines 163-166 - These two sentences feel a bit uninformative and out-of-place for a results section. Are they really necessary? I would consider rephrasing or even cutting these statements from the manuscript.

We omitted these two sentences in the revised version of the manuscript.

Line 196 - Add "log₂FC" - also check throughout to make sure of its presence.

We checked that \log_2 FC values are given wherever we explicitly refer to gene expression changes.

Line 198-199 - Wording makes it unclear which one of the two is lower.

We modified the sentence to make it clear that *lanM* was downregulated when comparing the Ln cocktail against Nd, **see lines 236-237**.

Line 225 - Say more about the nature of the type of t-test performed.

For the original manuscript we performed two-sample t-tests two compare between pairs of groups. However, given that we deal with three different groups it is more correct to use one-way ANOVA to avoid false positives. For the revised manuscript we tested the significance of the EM data using one-way ANOVA combined with Tukey-Kramer as post-hoc test. We added this information to the figure legend, **see lines 899-901**; the methods section, **see lines 552-554**; as well as the results section (**lines 273-274**)

Line 212 - Why is only one number reported for RHAL1_02672 when 2 numbers are reported for the other genes listed here?

RHAL1_02672 was only downregulated when comparing Nd against La. We clarified this in the revised manuscript, **see lines 253-254**.

Line 219 - Replace the "+" sign here.

Replaced as suggested, **see line 267**.

Line 233 - Replace the "+" sign here.

Replaced as suggested, **see line 280**.

Line 234 - How do you attribute distinct signals to specific lanthanides here? It is confusing.

The elemental composition of the observed lanthanide deposits was analyzed by means of energy-dispersive x-ray spectroscopy (EDX). EDX makes use of elements emitting characteristic x-rays upon x-ray excitation. We added this information to the main text, **see line 281**.

Line 236 - Use elemental abbreviations to be consistent throughout.

In the revised manuscript, we use elemental abbreviations throughout the manuscript. We added a list containing the members of the Ln series / rare-earth elements to the supplement, **see Table S1**.

Line 237 - Confused about why P is invoked here, there is no verbiage to preface why.

We provide more context in the revised manuscript. For the model methylotroph *M. extorquens* AM1 it was postulated that Ln are stored intracellularly in the cytoplasm in the form of Ln phosphates. The ratio between P and Ln can give us clues if lanthanides are complexed as phosphates or not, **see lines 291-295**.

Line 240-241 - Confusing way to start discussion section...

We rephrased the beginning of the discussion, **see lines 298-301**.

Line 251 - Suggested change to start the sentence: "Based on these data..."

Modified as suggested, **see line 320**.

Line 254 - A supplemental table of lanthanide names, since you are using so many of them, would be helpful to someone who does not regularly work with all these compounds.

We added such a table, please have a look at **Table S1**.

Line 257 - Insert text "in this study" in the sentence to reorient reader.

Modified as suggested, **see line 325**.

Line 261 - do not define metalation anywhere in the text. A little hint to the reader would be helpful.

Metallation is explained in the revised manuscript, **see lines 329-330**.

Line 274 - "in a non-methylotrophic background" alone is quite vague - could you use text to expand more on this a bit?

We explain this now in more detail, **see lines 343-345**.

Line 283-284 - Is this sentence important/necessary?

We omitted this sentence in the revised manuscript.

Line 286 - Is there something particularly interesting about pyruvate that makes it interesting/worth reporting on? Similarly, is the whole long following section focused upon Calcium completely necessary? The way the discussion is written, going back and forth between Ca and Ln is actually more confusing rather than informative.

Ln-dependent physiology is mostly studied in the context of methylotrophy and alcohol oxidation. We could previously (Wegner et al., 2021; <https://doi.org/10.1128/AEM.03144-20>) show that Ln supplementation has a positive effect on growth in a non-methylotrophic context, when Beijerinckiaaceae bacterium RH AL1 is grown with pyruvate as an alternative carbon source. Reasons for this effect remain elusive for now. Surprisingly, gene expression linked to the core lanthanome (e.g. *xoxF*, *lanM*) was responsive to Ln supplementation when strain RH

AL1 was grown with pyruvate, although the encoded proteins are not necessary for growth with pyruvate. Similar effects, to the best of our knowledge, have not been described for other microorganisms. Our findings suggest that lanthanides do affect metabolic aspects that are linked to/controlled by Ca. We revised the corresponding section in the discussion to make our reasoning more clear, **see lines 351-381**.

Line 297 - Typo.

The corresponding sentence has been rephrased, **see lines 359-360**.

Line 300 - What type of sensitivity are you talking about?

In the referenced study ([https://doi.org/10.1016/s0006-3495\(95\)79958-1](https://doi.org/10.1016/s0006-3495(95)79958-1)), Reusch and colleagues used bilayer patch-clamp techniques to study complexes of PHB and polyphosphate that form calcium channels. They found these channels to be sensitive to transition metals, and also lanthanides. We clarified this in the revised manuscript, **see lines 361-363**.

Line 313 - First mention of the term lanthanide phosphates. Should be broached earlier when P is first mentioned.

We explain our motivation to determine lanthanide phosphate ratios in the results section of the revised manuscript, **see lines 291-295**.

Line 348 - Is sulfur metabolism being invoked or not really? Seems like really soft language considering a lot of space in text and figures elsewhere have already been devoted to sulfur.

Our writing was misleading. The differential expression of genes linked to alkanesulfonate uptake/utilization was described as sign for oxidative stress in a couple of different species, and we believe this is also the case for Beijerinckia bacterium RH AL1. We revised the corresponding section in the discussion, **see lines 423-434**.

Line 355 - This is not necessary to be quoted or cited, in my opinion.

We do think that the phrase "lanthanide puzzle", which was used first in the referenced paper, captures our growing, but still limited understanding of lanthanide-dependent physiology very well.

Line 376 - Unclear if all samples were acclimated on 1) homologous conditions into which they were diluted, 2) if all were acclimated in identical condition, or 3) some hybrid of 1) and 2).

Please note our two answers in response to your comments methods (1) + (2).

Line 441 - Suggested rephrasing: "using the freehand selection tool in ImageJ..."

Modified as suggested, **see lines 548-549**.

Line 718-719 - Only 3 groups are listed here, list all 4 groups.

This was corrected in the revised legend of now Figure 5, **see lines 876-881**.

Line 719 - Add a period.

Modified as suggested.

Line 719 - this might be the only place in the paper where the lanthanome is directly defined. It should be defined in the text as well.

The term lanthanome is introduced upon the first usage in the main text, **see lines 218-224**.

Reviewer #2:

This paper by Gorniak et al., investigates how different lanthanides affect gene expression and metabolism using lanthanide-dependent, and lanthanide-accumulating methylotroph, Beijerinckiaceae bacterium RH AL1. Lanthanide accumulation in RH AL1 was previously shown by this group and in this work the authors looked at different lanthanides and concentrations of lanthanum, to identify the differentially expressed genes. Overall the work is exciting and the conclusions are valid. It is quite interesting to see the differences and commonalities in response to different Ln elements. The effects of differential gene expression and their physiological significance are yet to be understood.

Thank you for the positive feedback regarding our manuscript, we have addressed your suggestions in the revised version of the manuscript, please find our answers to your comments below.

1. Please include full forms of all abbreviations and details to the figure legends to help the reader understand the figures presented without referring back to the text in the results section. For instance in Fig. 1 - please mention La, Nd, cocktail stand for, panel A to the left shows lanthanum concentrations? Panel B is missing from the legend. In Panel B, what do copies refer to? Also, mention the number of replicates for the growth assays and experiments.

We have reworked all figure legends to increase clarity and readability, see for instance the modified legends of now **Figure 2** (formerly **Figure 1**) see **lines 842-853**, and **Figure 3** (formerly **Figure 2**) see **lines 855-869**. The legend of now **Figure 2** explains the Ln cocktail and also indicates the number of biological replicates. As we use a lot of different lanthanide elements, we followed a suggestion by reviewer #1 and added a table including all of them and some basic information to the supplement (**Table S1**). We also added the new **Figure 1** to explain the experimental setup in more detail.

2. Please mention the number of replicates performed for the RNA-seq experiment for the 4 different conditions (preferably in the main text and figure legend)

We added the number of replicates (n = 3) regarding the RNAseq experiment to the main text and figure legend, **please see the legend of Figure 2 and Figure 5, and lines 454 + 488**.

3. What do the terms "overrepresentation" and "gene enrichment" mean in terms of biology? Are all the genes that are identified from the gene enrichment analysis also identified as overrepresented in the overrepresentation analysis?

Overrepresentation analysis (ORA) is in essence a one-sided version of Fisher's Exact Test. Based on the available KEGG annotation of Beijerinckiaceae bacterium RH AL1 we know which genes belong to which pathway. Genes belonging to a pathway make up a certain proportion of all annotated genes of the genome. Imagine a genome comprises 1000 KEGG-annotated genes and there are 10 genes linked to glycolysis/gluconeogenesis (= 1% of all annotated genes). Overrepresentation analysis tests if genes associated with certain pathways are unexpectedly abundant in a list of genes. Let's say we have a list of 50 genes from the mentioned genome and this list includes all 10 glycolysis genes (10/50, 20%), in this case, glycolysis genes would be overrepresented. Gene set enrichment analysis is different from ORA and does not only rely on a list of genes, but it uses additional data, here changes in gene expression. Genes are ranked and it is tested if genes belonging to certain pathways are randomly distributed throughout the ranked list or if they occur at the top or bottom of the list. Please note that the modified figure no longer includes GSEA results, we derived our main findings from the ORA.

4. It is interesting to see hits in folate biosynthesis, secondary metabolism, two-component systems, etc but the gene expression appears different under La versus cocktail or pairwise comparisons. The authors could elaborate/discuss a bit more about the potential physiological implications of these changes in different pathways with respect to separate conditions in the discussion.

Thank you for bringing this up, we do not elaborate further on secondary metabolism, because secondary metabolism is a global KEGG pathway category that includes many central metabolic pathways (e.g. TCA cycle, glycolysis, pentose phosphate pathway). Single enriched genes were scattered over many pathways, which did not give us a clear readout. Folate biosynthesis is no longer mentioned since we now only show ORA results in

Figure 3 (formerly Figure 2). Two-component systems are also a rather broad KEGG pathway that overlaps with flagellar assembly and bacterial chemotaxis and includes genes from both. Genes can generally be assigned to multiple pathways. We see that this is misleading and clarify this in the results section, **see lines 158-159**. We mention now genes coding for the sensor kinases CheA and PleC as examples for distinct two-component system genes (**lines 159-161**) Chemotaxis and motility are covered in in the discussion, **see lines 350-359**.

September 22, 2023

Dr. Carl-Eric Wegner
Friedrich-Schiller-Universität Jena
Institute of Biodiversity, Aquatic Geomicrobiology
Dornburger Str. 159
Jena 07743
Germany

Re: Spectrum00867-23R1 (Different lanthanide elements induce strong gene expression changes in a lanthanide-accumulating methylotroph)

Dear Dr. Carl-Eric Wegner:

Thank you for submitting your manuscript to Microbiology Spectrum. As you will see your paper is very close to acceptance. Please modify the manuscript along the lines I have recommended. As these revisions are quite minor, I expect that you should be able to turn in the revised paper in less than 30 days, if not sooner. If your manuscript was reviewed, you will find the reviewers' comments below.

When submitting the revised version of your paper, please provide (1) point-by-point responses to the issues raised by the reviewers as file type "Response to Reviewers," not in your cover letter, and (2) a PDF file that indicates the changes from the original submission (by highlighting or underlining the changes) as file type "Marked Up Manuscript - For Review Only". Please use this link to submit your revised manuscript. Detailed instructions on submitting your revised paper are below.

Link Not Available

Sincerely,

Jannell Bazurto

Reviewer comments:

Please add the definition for methylotrophy to the abstract and amend Figure 3 such the numbers 1-4 in 2B and 2C are either replaced with specific conditions or appear in addition to the numbers.

Preparing Revision Guidelines

For complete guidelines on revision requirements, please see the journal Submission and Review Process requirements at

<https://journals.asm.org/journal/Spectrum/submission-review-process>. **Submissions of a paper that does not conform to Microbiology Spectrum guidelines will delay acceptance of your manuscript. "**

Please return the manuscript within 60 days; if you cannot complete the modification within this time period, please contact me. If you do not wish to modify the manuscript and prefer to submit it to another journal, please notify me of your decision immediately so that the manuscript may be formally withdrawn from consideration by Microbiology Spectrum.

Responses to reviewer comments:

Please add the definition for methylotrophy to the abstract and amend Figure 3 such the numbers 1-4 in 2B and 2C are either replaced with specific conditions or appear in addition to the numbers.

We modified Figure 3 accordingly, and added the definition for methylotrophy to the abstract.

September 25, 2023

Dr. Carl-Eric Wegner
Friedrich-Schiller-Universität Jena
Institute of Biodiversity, Aquatic Geomicrobiology
Dornburger Str. 159
Jena 07743
Germany

Re: Spectrum00867-23R2 (Different lanthanide elements induce strong gene expression changes in a lanthanide-accumulating methylophile)

Dear Dr. Carl-Eric Wegner:

Your manuscript has been accepted, and I am forwarding it to the ASM Journals Department for publication. You will be notified when your proofs are ready to be viewed.

Sincerely,

Jannell Bazurto
Editor, Microbiology Spectrum
